



# Measurement report: Influence of the Antarctic Ozone Hole in Southern Brazil: Conceptual model for 42 years of analysis the atmospheric dynamics on ozone

Gabriela Dornelles Bittencourt[1], Damaris Kirsch Pinheiro[2], Hassan Bencherif[3,5], Lucas Vaz Peres[4], Nelson Begue[3], José Valentin Bageston[1], Douglas Lima de Bem[2], Vagner Anabor[2], and Luiz Angelo Steffenel[6]

[1]National Institute for Space Research, INPE/COESU, Santa Maria, RS, Brazil
[2]Federal University of Santa Maria, Santa Maria, RS, Brazil
[3]Laboratoire de l'Atmosphère et des Cyclones, UMR 8105 CNRS, Université de la Réunion, Reunion Island, France
[4]Federal University of Western Pará, Santarém – PA, Brazil
[5]University of KwaZulu-Natal, School of Chemistry and Physics, Westville, Durban, South Africa
[6]University of Reims Champagne-Ardenne, LICIIS, Reims, France
**Correspondence:** Correspondence to: Gabriela Dornelles Bittencourt (gadornellesbittencourt@gmail.com)

**Abstract.** The austral spring in the Southern Hemisphere presents temporary reductions in ozone content mainly in the Antarctic region known as the Antarctic Ozone Hole (AOH). However, studies show an influence in mid-latitude regions, such as southern Brazil, where days with temporary decreases in the total column ozone (TCO) are identified. The main objective of this work is to investigate this influence of AOH on the southern region of Brazil, using data from the TCO and vertical profiles that will help to identify the preferential height at which these decreases occur in southern Brazil, in addition to analyzing the atmospheric dynamic behavior during these events in the period 42 years of data (1979 to 2020). The methodology used comprises the analysis of average daily data of the total column of ozone through ground-based instruments (Brewer Spectrophotometer), satellite data (TOMS and OMI), and to compare reanalysis data from the ECMWF-ERA5, for the identification of events of influence of the AOH on the southern region of Brazil. The analysis of the vertical content of $O_3$ data from the TIMED/SABER satellite provides daily data from 15 to 110 km in height and has 19 years of $O_3$ profiles available in the period from 2002 to 2020. From this, 102 events were identified that influenced Santa Maria (29.4ºS; 53.7ºW), in the south of Brazil, with a temporary decrease in the ozone content in the period, where between 24.1 - 28 km of altitude the more significant reductions in $O_3$ during events. In the dynamic analysis, the stratospheric fields showed an increase in the absolute potential vorticity, mainly in September and October. The conceptual models in the horizontal and vertical section of the atmosphere explain the action of the stratospheric and tropospheric jet during the occurrence of events of decrease in the $O_3$ content in Santa Maria. It was possible to identify the strong influence on the development of these events through the connection of the stratospheric jet (polar vortex) with the tropospheric jets (polar and subtropical jet) at medium and high levels of the atmosphere.



# 1 Introduction

The maintenance of life on Earth, through the energy balance of the planet, of all living beings, whether humans, animals, or plants, is due to the existence of ozone gas (Seinfeld and Pandis, 2016). In 1840, when scientists discovered the gas ozone ($O_3$), studies show that it is the most important trace gas for sustaining life on Earth, due to its ability to absorb ultraviolet (UV) radiation incident on the atmosphere. Ozone ($O_3$) is formed by photochemical processes in the atmosphere. Its highest concentration (about 90%) is found in the stratospheric layer, around 15 to 35 km altitude, in a region known as the "Ozone Layer" (London, 1985). Farman et al. (2003) detected a massive reduction in ozone content over the Antarctic region during the austral spring, which was later explained by a large-scale circulation known as the "Brewer Dobson Circulation" (BDC) (Brewer, 1949; Dobson et al., 1930; Butchart, 2014). This large-scale meridional circulation is responsible for transporting ozone formed in the tropics towards the poles and then to the troposphere in mid- and high-latitude regions (Butchart, 2014). This significant decrease identified by Farman et al. (1985) was called the "Antarctic Ozone Hole" (AOH) and is indicated in a region with total column ozone values (TCO) lower than 220 Dobson units (DU). The polar vortex acts as a barrier to the flow of air mass exchange between the AOH region and mid-latitudes. With the end of the polar night with the arrival of spring, the polar vortex becomes unstable due to the return of solar radiation and the increase in temperature in the region and increased activity of planetary waves. This instability allows poor ozone air masses to be ejected from the polar regions as filaments, reaching mid-latitudes (Manney et al., 1994; Stohl et al., 2003; Marchand et al., 2005) and southern South America (Kirchhoff et al., 1996; Bittencourt et al., 2018; Bresciani et al., 2018).

# 2 Methodology

The study area of this work comprises the southern region of Brazil in the city of Santa Maria (29.4ºS; 53.7ºW), according to figure 1. Total column ozone monitoring has 42 years of data available in Santa Maria, comprising ground-based instruments (Brewer Spectrophotometer between 1992 - 2017) and satellite data (TOMS, OMI during 1979 - 2020) to analyze the total ozone column $O_3$ (Vaz Peres et al., 2017), for the identification of the events of AOH influence in the region (Bittencourt et al., 2019). The database used include measurements available since 1979 with satellites and from 1992 ground-based measurements began to be carried with Brewer Spectrophotometers.

## 2.1 Ground-based data: Brewer Spectrophotometer

The Brewer spectrophotometer is one of the most important ground-based instruments that perform daily measurements of ultraviolet radiation, in addition to total column measurements of ozone ($O_3$) and sulfur oxide ($SO_2$). The measurements in the analyzed region began with the Brewer Spectrophotometer model MKIV #081 in the period from 1992 to 1999, while model MKII #056 operated from 2000 to 2002 and model MKIII #167 from 2002 to 2017, installed at the Space Observatory South – OES/CRS/INPE – MCTI. Brewer Spectrophotometers are composed of monochromators and detectors to observe and measure the spectrum of solar radiation, which consists of a scattering element and devices to control the width of the desired wavelength





band. The energy source (the Sun) must present a continuous spectrum and is formed by a spectrophotometer and a Sun tracking

system that are coupled to a microcomputer. In this way, the system performs the acquisition and storage of data and controls

the instruments through its own software Kipp Zonen Inc. (Sci-Tec. 1999). The individual $O_3$ observations obtained by the

Brewer Spectrophotometer use a method called: Direct Sun Measurement (DS). The instrument made 5 measurements for 3

minutes, using five wavelengths (306.3; 310.1; 313.5; 316.8; 320.1 $nm$) to infer TCO with a resolution of 0.5 $nm$. The average

DS measurement is validated if the standard deviation of the five ozone measurements is lower than 2.5 DU. This is made to

prevent most of the cloud interference in the data. The methodology employed by Brewer uses UV radiation measured at five

wavelengths (306 to 320 $nm$) and applies Beer's Law, which describes the attenuation of radiation by the components of the

atmosphere, to obtain the TCO. In this same analysis, it is necessary to use more than one wavelength to remove influences

from other gases, such as $SO_2$, and aerosols. At the end of each day, daily $O_3$ averages are calculated at the location where

TCO monitoring is carried out, and these same averages generated by Brewer's ground-based data are compared with satellite

measurements.

### 2.1.1    Satellite data: TOMS and OMI

The Total Ozone Mapping Spectrometer (TOMS) and the Ozone Monitoring Instrument (OMI) are the instruments on board

satellites that complement the database, when Brewer data was unavailable, for the analysis of the total column of ozone

in the study region of this work during the period from 1979 to 2020. This is possible because the data from the ground-

based instruments and data from the satellite-based instruments are in good agreement for the Southern Space Observatory

(Vaz Peres et al., 2017). The TOM'S instrument was one of the first satellite-based instruments with continuous observations

that were available for studies related to monitoring $O_3$ content with global and regional trends. The TOMS was developed

by the National Aeronautics and Space Agency (NASA) and began its activities in 1978, with the launch of the Nimbus-7

satellite. The instrument is pointed directly into the atmosphere (nadir view) to measure the amount of backscattered ultraviolet

(UV) radiation. TOMS has a fixed grid and a series of exit slits that scan the orbital band to within $\pm51°$ of nadir in $3°$

steps, with an instantaneous field of view (IFOV) of approximately 0.052 radians. At each instrument scan location, Earth

radiation is monitored at six wavelengths (310 and 380 $nm$) in the ultraviolet channel to infer total ozone. Between 1991

and 1994, the instrument was on board the Meteor-3 satellite. In 1996, it was replaced by the Earth Probe, and at the end

of 2005 the TOMS instruments ended its activities of measuring the TCO (Herman et al., 1996; McPeters et al., 1998). In

July 2004, the OMI instrument was launched on board the ERS-2 satellite in collaboration between the Dutch Agency for

Aerospace Programs (NIVR), the Finnish Meteorological Institute Agency (FMI) and the National Aeronautics and Space

Agency (NASA), which continues to provide TCO measurements to this day. OMI evolved from NASA's Total Ozone Mapping

Spectrometer (TOMS) instrument and the European Space Agency's (ESA) Global Ozone Monitoring Experiment (GOME)

(aboard the ERS-2 satellite). This new generation of satellites can measure more atmospheric constituents than TOMS and

provides much better ground resolution than GOME (13 km x 25 km for OMI versus 40 km x 320 km for GOME). The

OMI instrument was launched in July 2004 aboard the ERS -2 satellite and continued the recordings of the TOMS satellite,

which ended its activities in 2005. OMI measurements continue to this day, recording measurements of the TCO as well as





some atmospheric data related to $O_3$ chemistry, such as $NO_2$, $SO_2$, some types of aerosols, and cloud cover. The Earth is
observed in 740 bands along the satellite's orbit, large enough to provide global coverage in 14 orbits (1 day). The satellite
uses backscatter ultraviolet (BUV) technology with two images fed into a spectrometer grid and a spatial resolution of 13 x
25 km in the two UV bands: UV-1 (270 to 314 nm) and UV-2 (306 to 380 $nm$) with spectral resolutions of 0.45 and 1 $nm$,
respectively.

### 2.1.2 Satellite data: Vertical profile

The analysis performed here were based on measurements of the SABER instrument (Sounding of the Atmosphere using
Broadband Emission Radiometry) on board the TIMED (Thermosphere-Ionosphere-Mesosphere Energetics and Dynamics)
satellite (Russell III et al., 1999), which provides data from 2002 to the present day. The SABER instrument observes the Earth
in narrow spectral bands, where its observational coverage varies from 50°S – 82°N latitude to 82°S – 50°N every 60 days.
SABER evaluates atmospheric infrared emissions in the spectral wavelength range from 1.27 to 17 $\mu$m, where emissions of
carbon dioxide (15 $\mu m$), ozone (9.6 $\mu m$) and nitric oxide (5.3 $\mu m$) are studied. (Mlynczak et al., 1993; Russell III et al., 1999;
Dawkins et al., 2018). SABER is one of four experiments in NASA's TIMED mission, launched in May 2000 by a Delta II
rocket into a circular orbit at 74.1° $\pm$ 0.1° inclined, 625 $\pm$ 25 km where it scans ~10–105 km of altitudes for temperature and
between 15–110 km of altitudes for ozone every 58 s with approximately 96 sweeps per orbit and mapping about 14 longitudes
per day. In this work, version 2.0 of the SABER data was used, providing vertical ozone and temperature profile data, in the
period from 2002 to 2020 between 15 and 110 km of altitude. The study region was selected through a box of $\pm2$° latitude
and longitude from the reference point in Santa Maria (29.4ºS and 53.7ºW), and during the 19 years of data, approximately
6,715 daily vertical profiles were identified where the satellite mapped the selected region. The profiles were interpolated with
a vertical resolution of 0.1 km providing 954 altitude levels in the range of 15 to 110 km in height. The analysis of the $O_3$
vertical profile is extremely important for this work, mainly because it allows investigating the behavior of $O_3$ at different
heights, and at which heights the drop in $O_3$ content in the study region was greater.

### 2.1.3 Meteorological data: Reanalysis ECMWF-ERA5

Dynamic analysis of events includes the use of new generation meteorological data from the reanalysis of the European Center
for Medium-Term Weather Forecasts (ECMWF) with the new ERA5 (Hersbach et al., 2020). Data from the ERA5 reanalysis
were used here to study the dynamic behavior of the atmosphere during the occurrence of $O_3$ content decrease events, analyzed
by potential vorticity fields and vertical sections of the atmosphere. The ECMWF-ERA5 reanalysis has a horizontal resolution
of 0.25° x 0.25° latitude-longitude and a temporal resolution of 1 hour. Hoffmann et al. (2019) showed significant improvements
in potential temperature conservation, especially at stratospheric levels, when comparing data from the ERA5 reanalysis with
the ERA-Interim reanalysis (Bittencourt et al., 2019). In this study, daily data are used at 18 UTC at 37 isobaric levels (1000
to 1 hPa) during the period from 1979 to 2020. The domain region was comprised between 10 ºN to 90 ºS and 100ºW to 20ºE.
This area covers the south-central region of South America and part of the South Pole, showing in detail the behavior of the



advance of the poor $O_3$ air mass, after the definition of the event, described in the next section. The data used in this study were:

. Total column $O_3$ ($TCO_3$), for comparisons with TCO ground-based and satellite data over the SM.

. Potential vorticity, for analysis the stratospheric dynamics of AOH influence events.

. Temperature, zonal, meridional, and vertical components of the wind, for the vertical cuts in the identification of the jets (stratospheric and tropospheric).

Potential vorticity fields assist in monitoring stratospheric dynamic behavior. In addition, it allows identifying the origin of air masses poor in $O_3$, where in the Southern Hemisphere (SH) Absolute Potential Vorticity (APV) is used (Holton, 1995). Thus, when an increase in APV is observed, a polar origin of this air mass is identified, indicating an air mass coming directly from

the Antarctic region where AOH is active. When there is a decrease in APV, the origin is characterized as equatorial (Semane et al., 2006; Peres et al., 2016; Bittencourt et al., 2019; Bittencourt, 2022). After identifying the events (TCO) and determining the preferred height ( SABER data) at which these temporary decreases in $O_3$ content occurred, PV analysis was performed at height pressure levels of 20 hectopascal (hPa), which corresponds to the region of the low and middle stratosphere between 25 and 30 km of altitude. To identify the behavior of the jet streams during the active period of the AOH, fields with the vertical

cut of the atmosphere help in this matter. In this way, the vertical section of the atmosphere between 1000 and 5 hPa of potential temperature (in Kelvin) and wind (in m/s) for the longitude of 54° west, presents the behavior of the stratospheric jet (polar vortex) and the tropospheric jets (subtropical and polar) (Rodríguez et al., 2021).

## 2.2  Criterious for definition of influenced events of the Antarctic ozone hole

The daily average data obtained by satellite and ground-based instruments provided the analysis of 42 years of TCO data on

Santa Maria. To quantify the temporary reductions in $O_3$ content, tests were carried out and the analysis of the climatological mean -1.5 of its standard deviation was used, which showed that the use of -1.5 times the standard deviation ($\mu - 1.5\sigma$). These tests were carried out by Peres et al. (2016) who showed efficiency in the statistical criterion of -1.5, managing to identify events of high significance in the mid-latitude regions, being sufficient to quantify these temporary reductions (Peres et al., 2016; Wilks, 2006; Bittencourt, 2022). The first step is a climatological analysis of the TCO data, during the 42 years of $O_3$

monitoring, from August to November with days in which the TCO daily average ($TCO_d$) presented values lower than the monthly mean ($\overline{TCO_m}$) minus 1.5 times the standard deviation of the month ($\sigma_m$) are selected for stratospheric analysis.

$$TCO_d < \overline{TCO_m} - 1.5\sigma_m \tag{1}$$

The second step after identifying these events is the analysis of potential vorticity fields. Reductions the $O_3$ content are presented in percent (%) and in absolute values (DU).





### 2.2.1 Vertical profile analysis of TIMED/SABER $O_3$ satellite

After identifying the events on 42 years of data, the SABER satellite analysis is used to characterize the vertical behavior of the atmosphere during the events of influence of the AOH on the region. The analysis was generated from 19 years of data available on the platform, where the daily behavior of the data was observed through vertical profiles, identifying the height where the greatest drops in ozone content happen during the occurrence of an event. The climatology and monthly averages were generated, so it was possible to make comparisons of the events with the monthly climatology. To quantify the drop in $O_3$ that occurs during the event, the calculation of the relative differences (in percentage) helps to show this temporary decrease in numbers. The calculation of the difference is based on the difference between the profile of the day of the event and the climatology profile for the reference month (from August to November - spring months), following equation 2:

$$RD(\%) = 100 * \frac{(EventDay - ClimatologicalMonthh)}{(ClimatologicalMonthh)} \tag{2}$$

## 3 Results and Discussion

### 3.1 TCO Climatology in Santa Maria

Monthly climatology and standard deviation are shown in Figure 2 for the four main TCO measurement instruments at the subtropical station in SM from 1979 to 2020, where we have Brewer (black), TOMS (blue), OMI (red) and reanalysis of the new version of ECMWF-ERA5 (green). The annual variability stands out in the TCO data in Santa Maria, with minimum values in autumn between 255 and 260 DU, and maximum values during spring with values between 295 and 300 DU (Vaz Peres et al., 2017; Bittencourt, 2022), and this variability is mainly explained by the large-scale motion known as the BDC (Brewer, 1949; Dobson, 1968). This transport is the dominant process that determines, through its meridional movement, that the $O_3$ produced in low latitudes is transported to medium and high latitudes, causing this maximum to occur in late winter/early spring (London, 1985). Statistical analysis between databases found interesting results comparing ground-based instruments versus satellites/reanalysis. Comparisons were separated by instruments based on the available data series, with Brewer vs. TOMS for the period June 1992 to December 2005, contains 2164 pairs of data, Brewer vs. OMI between October 2004 to December 2017, has 4621 pairs of data, and to finish off Brewer vs ERA5 for January 1979 to December 2020, with about 5065 data pairs. observed that the correlation coefficient (R2) with respect to the instruments presented considerably good values, where the values of the correlation coefficient were: 0.88 (BREWER vs TOMS) and 0.94 (BREWER vs OMI). Updates to instruments onboard satellites over the years may explain these differences between TOMS and OMI. The exception was the set between BREWER and ERA5 where R2 presented a value around 0.83. Despite not being such a low value, the use of TCO reanalysis data cannot represent the behavior of $O_3$ with a good quality. Regarding data from TOMS and OMI satellites, previous studies (Antón et al., 2009; Toihir et al., 2015; Vaz Peres et al., 2017) have shown similar results in relation to comparisons between these TCO measurement instruments over different regions. Dynamic influences at subtropical latitudes modify $O_3$ behavior throughout the year, unlike what happens in other regions (Reboita et al., 2010). The presence of jet streams at high levels of the atmosphere helps transport air masses, especially when observing stratosphere-troposphere exchanges, where the position





and speed of these jets (subtropical or polar) as a function of the meridional temperature gradient can determine the variation of ozone in the atmosphere (Bukin et al., 2011). Another important point of the subtropical site is that AOH indirectly influences the behavior of $O_3$ mainly during the austral spring (Bittencourt, 2022). Sivakumar et al. (2007) carried out a study on the climatology and stratospheric ozone variability over the Ile de La Réunion, France, for 15 years of data available in satellite instruments (HALOE, SAGE-II, TOMS), ozonesondes. In this work, the authors also identified maximum $O_3$ values in spring and minimum values in autumn. (Santos et al., 2016) showed a high number of stratosphere-troposphere exchanges (STE) identified during the winter and spring months, with a lower frequency during the summer, also explained by the large-scale circulation (BDC).

### 3.1.1 Influence Events of the Antarctic Ozone Hole

The identification of the events of influence of the Antarctic Ozone Hole on the SM region follows the methodology previously presented, where first the daily average data of the 42 available years are analyzed and the climatological averages between August and November and their respective standard deviations are calculated. Then, the limiting values of the climatological mean minus 1.5 times the standard deviation ($\mu - 1.5\sigma$) are calculated for each month. Finally, the identification and confirmation of the event is based on the analysis of potential vorticity fields to verify the origin of this poor $O_3$ air mass, whether it comes from the polar region or the equatorial region. Table 1 presents the monthly climatology to identify the days when there is a decrease in the $O_3$ content during the occurrence of AOH. Possible events are selected based on the analysis of the average daily value of $O_3$, which must be less than the limit for the month of the event ($\mu - 1.5\sigma$). With this methodology, around 5124 days were analyzed during the study period (1979-2020). From this total number of available days in the 4 months, 735 days were selected because they had a daily TCO value below the -1.5$\sigma$ limit. Of this total, 102 AOH influence events were identified influencing the study region, Santa Maria, in the period from 1979 to 2020. (Bittencourt et al., 2019) showed, in 11 years of data, 37 AOH influence events over the southern region of Brazil from 2006 to 2017, in addition to presenting the dynamic stratospheric and tropospheric behavior during the occurrence of these events over the course of region.

## 3.2 Case Study

### 3.2.1 Event 20 October 2016

The AOH secondary effect event that occurred on October 20th, 2016, in Southern Brazil was one of the most intense ever recorded in the last two decades (Bittencourt et al., 2018; Bresciani et al., 2018), after Kirchhoff et al. (1996) identified one of the first extreme events of secondary effect of AOH on the southern region of Brazil that occurred on October 28th, 1993. (Bresciani et al., 2018) performed a multi-instrumental analysis of this event, using vertical profile data with satellite instruments (AURA and SABER) and ozonesondes, and showed how much this event impacted the region with an extreme temporary decrease in $O_3$ content. This study analyzed vertical profile data, through the launch of ozonesondes that identified this decrease in TCO in the region, in addition to ground-based instrument measurements (BREWER) and TOMS satellite data. In addition, a complete analysis of this 2016 extreme event in relation to the case study atmospheric dynamics analysis is





presented, using the climatology of 42 years of TCO data, SABER vertical profile data, and reanalysis of the ECMWF-ERA5
in the dynamics analysis. of the atmosphere, potential vorticity at 20 hPa and vertical section of the atmosphere (1000 at 5
hPa). The first part the analysis of the TCO data provided by the ground-based and satellite instruments. For this event, TCO
registered a minimum value on the October 20th. Brewer Spectrophotometer recorded in the TCO of the day a value of 225.5
DU, representing a reduction of around 23% in the ozone content in relation to the climatological average for October which
was $291.4 \pm 8.2$ DU. On October 21st, the secondary event continued to influence the region, where 233 DU was recorded,
resulting in a reduction of around 20% in relation to October's climatology. Figure 3 shows the average daily TCO values for
October 2020, with the values from the Brewer (blue), OMI (red), and ERA5 (green). The black line represents the limit $- 1.5\sigma$
of the standard deviation for the month, which according to Table 1 was 271.3 DU. The yellow circle represents the day of
the event on which the BREWER recorded the lowest $O_3$ content value. Blank spaces represent lack of data from both study
instruments. The beginning of October, according to Figure 3, presents TCO values ranging from 300 to 320 DU in the first
10 days of the month. Between the 14th and 15th, it is possible to identify that the TCO begins to show a drop in relation to
the limit of the climatological average, which despite showing flaws in the data, identifies a recurring decrease in $O_3$, where
on October 20th it presents its greatest decrease of ozone content for this month. Temporary decreases in $O_3$ content were
recurrent, and according to the data they influenced the region until at least October 23rd, 2016 (Bittencourt et al., 2018, 2019).
The October 2016 event was also identified with data from the $O_3$ vertical profile by SABER (Bresciani et al., 2018). Figure
4 shows the vertical $O_3$ profile of October 20th, 2016 (red) compared to the climatology of the event month (black). This
climatology was performed for the 19 years of data from the SABER satellite, allowing a more in-depth analysis of the event
in relation to the vertical behavior of the $O_3$ content. Figure 4a shows the vertical profile in $O_3$ partial pressure (mPa) data for
the day of the event (in red line), October 20th, 2016, and the climatology (black line) for the SABER data analysis period
(2002-2020). The region between 22 to 29 km in height, highlights the significant reduction that occurred during the event
in relation to climatology, ranging from 100 to 60 (mPa) with a peak at 24 km. (Bresciani et al., 2018) presented a multi-
instrumental analysis of the event identified in October 2016. The results showed the decrease in $O_3$ content according to data
from different vertical profile instruments, during the occurrence of this extreme event, and showing this reduction in the $O_3$
content. This reduction is observed when calculating the percentage differences for the day of the event, as shown in Figure
4b. The stratospheric region between 22 - 30 km of altitude, presents high values of reduction of the $O_3$ content in the region
during the event which was around -43% in relation to the October climatology by the SABER satellite.

### 3.3 Dynamic Event Analysis

The confirmation of the secondary effect event and the origin of the ozone-poor air mass is the dynamic analysis of this event,
through stratospheric dynamics using potential vorticity on isentropic surfaces. Figure 5 shows daily monitoring of PV at 20
hPa height at pressure levels, using ECMWF-ERA5 reanalysis data for two previous days (18 and 19 October 2016), on the
day when the lowest $O_3$ content was observed (October 20, 2016) and one day after the event (October 21, 2016). The increase
in the absolute potential vorticity over the region is evident in the sequence of days, where the arrival of an air mass poor
in $O_3$ from the polar regions to the mid-latitude regions is observed, identified by the increase in the APV, in the vorticity





fields of stratospheric potential showing that the origin of this poor $O_3$ air mass was polar. On October 18, 2016 (Fig. 5a) high APV values were observed advancing towards southern Brazil, with values ranging from 60 to 140 between the extreme south of Argentina and Uruguay. On October 19, 2016, (Fig. 5b) the southern region of Brazil was already showing signs of influence of the poor $O_3$ air mass, with an increase in APV, ranging from 80 to 120 potential vorticity units (PVU). On day 20 (Fig. 5c), the poor $O_3$ air mass advanced under the region, with an intensification of PVU values, ranging from 160 to 200 PVU over SM. Also noteworthy is a waveform advancing over the region as the poor $O_3$ air mass moved over Brazil. On the following day, October 21, 2016 (Figure 5b) there is a stabilization of this mass, but with a decrease in PVU values compared to previous days, around 160 to 180 PVU in the region. Bittencourt et al. (2019) presented an analysis of 11 years of TCO data, where AOH influence events on the SM region were investigated using ECMWF ERA-INTERIM data in the dynamic study of events, showing the secondary effect events tend to act in the region up to at least 3 days after the event itself is registered. (Peres et al., 2014) presented results that show the identification of events in the region for the year 2012, using ground-based and satellite data, and reanalysis data from NCEP/NCAR to monitor the dynamics of the stratosphere. Two events were identified influencing the region (Peres et al., 2014). The decrease in $O_3$ content during the event identified in October 2016 was detailed by Bittencourt et al. (2018) where, through the MIMOSA model (Modélisation Isentrope du transport Mésoéchelle de l'Ozone Stratosphérique par Advection), it was possible to identify the advance of a polar poor $O_3$ air mass over mid-latitudes regions. MIMOSA is a high-resolution model developed by the Service d'Aéronomie within the framework of the European METRO project (MERidional TRansport of Ozone in the lower stratosphere) (Hauchecorne et al., 2002). The results found by Bittencourt et al. (2018) with the MIMOSA model showed the direct impact of the ozone hole in southern Brazil at a potential temperature of 550 Kelvin and an altitude of approximately 22 km, and it is possible to identify the advance of a poor $O_3$ air mass over regions of medium latitudes. Bittencourt et al. (2018) identified the advance of the October 16, 2016, passing through the Antarctic region where AOH is acting and taking this air mass to mid-latitude regions through analysis of the retroactive trajectory by the HYSPLIT/NOAA model of this air poor in $O_3$. And this trajectory was identified at three different heights, 26 km, 24 km, and 22 km altitude at 18 UTC on October 20, 2016. Bittencourt, 2018 identified, during 11 years of analyzed data, 68% of the cases found occurred after the passage of frontal systems, and about 91% presented stratospheric and/or polar jet streams acting together with the events (Bittencourt et al., 2019). Canziani et al. (2002) showed that the performance of synoptic-scale systems can develop together, that is, in the stratosphere and in the troposphere, also known as Cut-off Lows (Pinheiro et al., 2021), favor the occurrence of Influence events of the Antarctic Ozone Hole over southern Brazil. The presence of stratospheric (polar vortex) and tropospheric (subtropical and polar) jets can be inferred in Figure 6 between October 18 and 21, 2016. The presence of the stratospheric jet, or polar vortex, is identified on October 18, 2016 (Figure 6a) between 70 and 80 ºS with moderate intensity ( 50 m/s) above 50 hPa. Below 200 hPa, the weak polar jet is observed acting in the most polar regions, while the subtropical jet stands out between latitudes of 30 - 40 ºS coupled to the polar vortex (above 100 hPa). On October 19th (Figure 6b) this stratospheric jet moved eastwards, still coupled to the polar jet, but now between 55 - 70 ºS. At high atmospheric levels (between 200 hPa) the coupling between the jets is visible, and its influence reaches the mid-latitude regions analyzed in this work ( 30ºS). With moderate intensity, the polar and subtropical jets extend to the middle troposphere around 500 hPa reaching up to 600 hPa. On the day of the event, October 20, 2016 (Figure **??**) and one day after the event





(Figure 6d), the influence of the polar vortex reaches high levels of the atmosphere, in addition to observing a coupling between the stratospheric jet and the polar jet that intensifies, between 50 - 60 ºS. On October 21, the coupling between them reaches

700 hPa, and the subtropical jet also shows an intensification on these two days. These large-scale systems help to understand that they have a direct influence on this movement of air masses between atmospheric layers. Previous studies show that this exchange between stratosphere and troposphere during the occurrence of AOH influences events on the region of study usually occurs after the passage of frontal systems so that the presence of jets helps this exchange of air masses between the layers Santos et al. (2016); Bittencourt et al. (2018, 2019).

### 3.4  Events Statistics

In this section, results obtained in relation to the main statistical analysis of the behavior of $O_3$ are presented, through the potential vorticity fields, analysis of the vertical profiles identified during the events (by the SABER satellite) and the dynamics of the jets.

#### 3.4.1  TCO statistics event

During the 42 years, the identification of AOH side effect events in the SM region occurred through the analysis of average daily data available with different $O_3$ measurement instruments already described here. Table 2 presents a summary of all events that were identified in the region through the methodology described above, where the day of the event and the reduction of $O_3$ in percentage (%). During the austral spring (August to November) more than 5124 days were analyzed during the study period (1979-2020). Among this total number of days available in the 4 months, about 735 days were selected because they had a daily

TCO value below the limit of $\mu - 1.5\sigma$ presented in Table 1. Of this total, 102 AOH influence events were identified influencing the study region, Santa Maria, in the period from 1979 to 2020. Peres et al. (2019) presented the climatological analysis for 35 years of data, 1979 to 2013, for the same region where they identified about 62 events that affected the region. Bittencourt et al. (2019) showed, in 11 years of data, 37 events of influence of AOH over the southern region of Brazil from 2006 to 2017, in addition to presenting the dynamic stratospheric and tropospheric behavior during the occurrence of these events over the

region. Table 2 also presents the phase in which the Quasi-Biennial Oscillation (QBO) was found during the event. According to some studies (Vaz Peres et al., 2017; Toihir et al., 2015, 2018) on the analysis of $O_3$ content variability, it is highlighted that the influence of QBO is more significant in tropical regions than in subtropical latitudes such as SM. Toihir et al. (2018) also identified that the most important variability in the $O_3$ content are, in addition to the annual oscillations, the zonal modulation of the wind at 30 hPa with the QBO, where it was possible to observe that the variability of the predominant $O_3$ content is

modulated in a cycle of approximately 2 years linked to the QBO, where tropical latitudes are in phase with the QBO, while regions in subtropical latitudes show an antiphase (Bittencourt, 2022). The importance of studying the behavior of the QBO under subtropical and tropical latitudes is due to its influence on the modulation of temperature in the stratosphere, affecting the photochemical dynamics in the lower stratosphere, and consequently influencing the behavior of large-scale circulation (BDC) throughout the atmosphere, which distributes the $O_3$ content from the formation regions (in the tropics) to regions of mid and

high latitudes (Cordero et al., 2012). The mean reduction, in percentage, of TCO in the 102 AOH influence events identified





in the study region during the 42 years of analysis was $8.3 \pm 3.5\%$. These events showed that September and October, months in which AOH is at its maximum activity peak, were the months in which the greatest number of temporary decreases in $O_3$ occurred due to these secondary influences. In the period 1979-2020, during the austral spring, about 32.3% and 37.2% of the events identified were in September (33 events) and October (38 events). Meanwhile, in August, when the polar vortex begins

to lose its intensity due to the end of winter and the return of radiation, 22 (21.5%) events influenced the study region, while in November only 9 (8.8%) events were registered. These results agree with (Bittencourt et al., 2019) where they identified, during 11 years of analysis of AOH influence events on the mid-latitude region, that September and October characterize the months with the highest occurrence of these events, when AOH reaches its maximum peak (Solomon, 1999; Vaz Peres et al., 2017; Bittencourt et al., 2019). According to Table 2, during the 42 years of analysis, 38 years registered at least one event of

influence of AOH on MS, totaling the 102 events identified. During the 38 years of events, 18 years the QBO was in its positive phase, and about 20 years in the negative phase of the QBO. In the positive phase of the oscillation, 61 events (59.8%) were identified, while 41 events (40.2%) occurred in the negative phase of the oscillation. Despite this, the years of positive QBO presented a greater number of events per year (on average about 3.22 events per year, while in the negative phase 2.14 events per year), which indicates that the positive phase of QBO is the most favorable for the occurrence of multiple annual events of

AOH influence over mid-latitudes. The causes of these QBO phase differences would need to be further investigated, and more studies should be performed to explain why most events occur during the positive QBO phase in the study region.

### 3.4.2   Vertical Profile Statistics

After identifying the AOH influence events in the period 1979 - 2020 with ground-based and satellite instruments (Table 2), it was possible to study the vertical profiles of the SABER satellite for the period of 19 years in which the satellite maintained

its activities, to identify the predominant altitudes at which these AOH-influenced events occur over SM. During this period (2002-2020) 43 vertical profiles of AOH influence events over the study region were found, according to satellite availability. Figure 7 presents a histogram with the frequency of occurrence of events according to the altitudes defined for this analysis, starting the verification at 20 km of altitude, where they were separated into five groups of altitude. According to the histogram, the predominance of events is in the months of September (31.6%) and in October (37%), August and November continue to

be the months with the lowest occurrence of event identification, 18.4% and 13.2%, corroborating the analysis made with TCO data. The heights that predominate the temporary decreases in $O_3$ content during the AOH influence events are the height between 24.1 - 28 km, followed by the lowest layer of the stratosphere between 20 - 24 km of altitude, in the months of greater occurrence of the events. This most intermediate layer of the stratosphere comprises the $O_3$ layer region, thus with active AOH, and influencing the mid-latitudes regions with the advance of poor $O_3$ air masses explains these greater decreases in $O_3$ content

at these heights. Dynamic influences that alter the behavior of $O_3$ in the lower stratosphere explain these decreases at these altitudes. In addition, $O_3$ at altitudes between 32 to 40 km shows changes, which can be explained by the residence time of the ozone molecule being shorter at higher altitudes due to photochemical influences, causing the air mass in the upper stratosphere to recover the ozone more quickly due to the photochemical production of the molecule at this altitude. From these results, it was established to analyze the events at altitudes of 24.1 - 28 km, due to the predominance of decreases in the $O_3$ content





at these altitudes of the events of influence of AOH on SM, by the analyzes of the SABER vertical profile. Figure 8 presents vertical profiles relative to the average of events in each month with their respective monthly climatology, in units of partial pressure (mPa) of $O_3$ at the altitude of 24.1 to 28 km. Regarding climatology, the months of September and October also stand out here, showing more intense reductions than their climatology, around 15-25 mPa of reduction. The month of November presents a series of factors that must be considered: the AOH begins to close in that month, recovering the ozone content, due to the weakening of the polar vortex and the increase in temperatures. Thus, this recovery begins to be observed in the average profile in relation to climatology during the period. Figure 9 shows the percentage differences for the 43 vertical profiles of events identified in the SABER analysis period (2002 to 2020) within the range of 24.1 to 28 km altitude. The differences found are around 20% for the months of August, September, and October, while the month of November shows, on average, differences around 15%. In agreement with previous studies (Peres et al., 2016; Bittencourt et al., 2019), these results show that most events occurred in the months of peak AOH (September -14 events and October - 15 events) and a smaller number of events in the beginning (August - 8 events) and end (November - 6 events) of AOH.

### 3.4.3 Dynamic Statistics Events

The stratospheric dynamics during the period of influence of AOH on SM presents important characteristics. The activity of the Antarctic polar vortex controls the behavior of the Ozone Hole, where a gradual increase in absolute potential vorticity is observed, and this gradual increase in the first months when the AOH is intensifying (August and early September) can be explained by the dynamics of the Antarctic polar vortex, which reaches its maximum intensity between late winter and early spring, when sunlight returns and destabilizes this vortex (Bittencourt, 2022). The return of sunlight, and the consequent destabilization of the polar vortex with the increase in temperature, causes the poor $O_3$ air that is trapped inside this vortex to move to other regions. The stratospheric field anomaly is shown in Figure 10 at 20 hPa in the middle stratosphere region. The calculation considers the events of each month identified in this work (Table 2). Negative PV anomalies act over the entire mid-latitude region of southern Brazil, including SM. The months of September and October stand out with the highest number of AOH-influenced events, Figure 10b and 10c where values range from -5 to -20 $mKs^{-2}hPa^{-1}$, reaching -40 $mKs^{-2}hPa^{-1}$ in October. Of the 102 events identified between 1979-2020 and the climatology of the period, the event anomaly in Figure 11 stands out, where a wave format with a predominance of negative anomalies is observed over the study region. Negative anomaly values range from -15 to -25 PVU (figure 11), and this analysis is in line with the results presented by Peres et al. (2019) who analyzed the TCO for 35 years for SM using data from the II NCEP/DOE reanalysis in the potential vorticity analysis, where they identified a wave pattern over the southern region of Brazil during events of decrease in $O_3$ content. The dynamic behavior of the atmosphere during AOH-influenced events in mid-latitude regions is of paramount importance to understand how these events manage to advance by modifying the $O_3$ content. Vertical analysis of the atmosphere shows us how jet streams, important large-scale systems, help transport air masses between the two atmospheric layers (stratosphere-troposphere) (Bukin et al., 2011; Bittencourt et al., 2019). According to the events identified in the 42 years, Figure 12a,12b,12c e 12d presents the climatological average of the jet streams with the wind components in m/s (purple shading) and, potential temperature (solid black line) in Kelvin, of the events per month (August to September) in the vertical section of the atmosphere




between 1000 and 5 hPa. The influence of the stratospheric jet, or polar vortex, stands out at higher levels of the atmosphere
(100 – 5 hPa) and at high latitudes (45ºS – 70ºS), according to figure 12. As previously described, August and September present
an extremely strong and very active vortex, which is explained by the low temperatures that are stored inside the vortex during
the polar night, making it a stable vortex during this period. The coupling between the jets together with the funneling of the
isentropic ones physically explains how the stratosphere-troposphere exchanges occur. (Santos et al., 2016) presents important
results regarding air exchange between the stratosphere and troposphere, noting the importance of the jet stream helping this
exchange mechanism, being more intense in the winter and spring months over Santa Maria region. (Rodríguez et al., 2021)
identified that the position and intensity of the subtropical jet directly affects the position of the dynamic tropopause, which
ends up influencing the mechanism of air mass exchanges by increasing/decreasing the amount of $O_3$ at stratospheric levels.
The coupling between the stratospheric and tropospheric jets is visible, highlighting a greater intensity of these jets compared
to their climatology. This pattern is confirmed in the average of all events identified in this work (102) during the 1979 to 2020,
presented in Figure 13. The coupling between the jets shown an important dynamic pattern during the occurrence of $O_3$ content
decrease events over the southern region of Brazil.

## 4    Conceptual model

The elaboration of the conceptual models presented in figure 14, allowed a better visualization of how the atmospheric dy-
namics behaves during the occurrence of events of influence of the Antarctic Ozone Hole in regions of medium latitudes over
South America, such as the city of Santa Maria (Bittencourt et al., 2019). Figure 14a shows the meridional/horizontal behavior
of stratospheric and tropospheric jets during the analysis period of this study (1979-2020), which is the active AOH period. At
20 hPa is the pressure level where the core of the stratospheric jet, or polar night jet, is intensely present during the winter/early
spring period. During the winter, the polar stratospheric region presents very negative temperatures inside the vortex, mainly
due to the absence of solar radiation. The polar stratospheric clouds that form in this period serve as the basis for the destruc-
tion of the $O_3$ molecule during this period (Solomon, 1999). As the climatology shows during the events of decrease in the
$O_3$ content, the polar jet is persistent and intense until the beginning of October, being able to influence up to 50 ºS of latitude
in high levels. At medium atmospheric levels, around 200 hPa, the position of the tropospheric jet follows the climatological
pattern. The subtropical jet mainly influencing the middle latitudes, and the polar jet being more important in the high latitudes
helping to transport air masses. The isentropic coupling is noticeable in the analyses, dynamically explaining the movement
of air masses from the stratosphere to the troposphere. The results also showed that stratospheric and tropospheric jets couple
during the occurrence of events, showing that these transfers of air masses between layers occur with the help of these large
dynamic systems, such as jets. Conceptually, Figure 14b symbolizes the dynamic behavior of the atmosphere during the 42
years of data analyzed over SM in relation to the identification of events of decrease in $O_3$ content, in the active period of the
Antarctic Ozone Hole, which occurs from August to November. Shades in purple show the jet intensity (in m/s) of both the
stratospheric and the two tropospheric jets (polar between 70ºS - 50ºS, and subtropical between 40ºS to 20ºS). The blue line
represents the movement of poor $O_3$ air mass from the pole region where AOH is active to mid-latitude regions, such as the city





of Santa Maria. The connection of the jets helps in this transport, causing a funneling of the isentropic, also affecting the height of the tropopause, causing this air with little $O_3$ to reach these regions. As shown in Figure 14a, b, air parcels with low $O_3$ concentration are "closed" within the polar vortex. Despite being stable, the polar vortex around the Antarctic Ozone Hole is

very dynamic and, therefore, air masses can "let go" of this vortex reaching regions of medium latitudes (Semane et al., 2006). This release occurs, as shown by previous works Bittencourt et al. (2019) after the passage of frontal systems and, according to the analysis presented in this paper, the coupling between the layers occurs through the connection between the subtropical and polar jets with the stratospheric jet. This coupling between the jets during these events occur in practically 90% of all events identified in analysis by (Bittencourt et al., 2019). The connection between the jets dynamically explains how the parcels of air

with less $O_3$ that are found in the upper levels of the atmosphere, in the stratosphere region and in high latitudes, reach regions of medium latitudes.

## 5   Conclusions

This work presents an analysis with average daily data of the TCO over the city of Santa Maria, a region of medium latitudes during the occurrence of events influenced by AOH. Ozone behavior over these regions relied on data from ground-based

instruments, such as the Brewer Spectrophotometer (1992-2017), and satellite instruments (TOMS and OMI), and TCO reanalysis (1979-2020) from ECMWF-ERA5 providing measurements for the 42 years (1979–2020). Vertical $O_3$ profiles provide a more detailed analysis of the atmosphere as altitude increases, so to identify the predominant altitudes where poor $O_3$ air masses reach mid-latitude regions, TIMED/SABER satellite data provide daily measures for 19 years of analysis (2002-2020). The TCO climatology shows a well-defined seasonal variation in the 42 years of analysis over the southern region of Brazil,

with maximum values during spring and minimum values during autumn (Vaz Peres et al., 2017; Bittencourt et al., 2019). This variation is mainly explained by the large-scale circulation (BDC) that transports air rich in $O_3$ from its region of formation in the tropical stratosphere to the regions of medium and high latitudes, at the poles. This transport dynamically explains the high $O_3$ concentrations in spring in the Santa Maria region. The methodology for identification of AOH influence events in the study region was able to find 102 events that decreased the $O_3$ content in the region between 1979-2020. The case study presented in

the results is an extreme $O_3$ decrease event in southern Brazil, which occurred in October 2016. According to TCO data, the value on the day of the event was 225 DU, reducing about 23% of the $O_3$ content, in relation to the climatological average for the month of October in SM. On the day of the event, the analysis of the vertical profile identified reductions of around -40% between 23 and 26 km of altitude, in relation to the climatology for the data period (2002-2020). The reductions observed in the vertical profile on the day of the event showed that most of the stratospheric layer, up to at least the middle stratosphere,

presented significant reductions in $O_3$ content over Santa Maria. For atmospheric dynamics analysis, the stratospheric potential vorticity fields identified at 20 hPa a significant increase in absolute potential vorticity over the region between October 19th and October 21st. Statistical analysis showed that, regarding TCO, the average reductions in AOH influenced events are around $8.3 \pm 3.5$ %. Most of the identified events occurred in September with 33 events (32.3%) and in October with 38 identified events (37.2%). This result agrees with the analysis of the vertical profiles used by the SABER satellite between 2002-2020,





where 43 event profiles were identified. The events profiles showed the greatest reductions in total $O_3$ content in September and October. Furthermore, the preferred height at which these reductions were found was in the layer between 24.1 and 28.0 km in most events. For future work, longer study periods will be analyzed to characterize and identify a greater number of AOH influenced events in mid-latitude regions, using other vertical profile databases. For the dynamic analysis of the stratospheric potential vorticity fields, data from the ECMWF-ERA5 reanalysis were used to create the potential vorticity fields to identify

the origin of the $O_3$-poor air mass during AOH-influenced events, where an increase Absolute potential vorticity (APV) indicates the polar origin of the air mass. According to the case study presented here, a significant increase in APV is observed during the 19th and 20th of October, indicating a polar origin of the air mass. In the statistical analysis of the stratospheric dynamics, the PV fields during the months of AOH influence in SM, showed negative PV anomalies over the region, where the highest values found are in the months of September and October, mainly. The coupling of stratospheric and tropospheric

jets had a strong influence on this study. The vertical section of the atmosphere from 1000 to 5 hPa presented in climatology a pattern consistent with the behavior of the polar vortex, as already mentioned, it is more intense and stable until the end of the polar night. Throughout this period, the subtropical tropospheric jet remains connected with the polar vortex, explaining this mechanism of air mass exchange between stratosphere-troposphere during the study period. The average during AOH influence events over southern Brazil characterizes this strong connection between stratospheric and tropospheric jets during

the analyzed period. The results of this work showed the occurrence of AOH influence events in the Santa Maria region in 42 years of analysis, where a maximum decrease in $O_3$ content was observed during these events at altitudes between 24.1 - 28.0 km of altitude according to the vertical profiles, and predominance of events in the months of September and October. Another point to be highlighted is the importance of the dynamic behavior of the atmosphere, through stratospheric and tropospheric jets during the events. The coupling between the jets, as seen in the data and transport to the conceptual model presented,

evidence, and explains how the exchanges between the atmospheric layers (stratosphere and troposphere) behave during the events of AOH influence of lower latitudes. A well-characterized isentropic funneling is observed over the region of the equatorial entry of the subtropical jet, highlighting this influence between the jets during the occurrence of $O_3$ content decrease events in mid-latitudes.

*Data availability.* The ozone ground-based data obtained by the Brewer spectrophotometer are available online at WOUDC station ID 529

(https://woudc.org/data/stations/index.php?lang=en). More information about these data can be obtained by contacting the corresponding author or, alternatively, José Valentin Bageston (jose.bageston@inpe.br). The TOMS and OMI-ERS2 satellite data are available at https://ozoneaq.gsfc.nasa.gov/data/toms/ and https://aura.gsfc.nasa.gov/omi.html. The vertical profile TIMED/SABER satellite data used in this work was downloaded using the SABER Custom Data Services tool available at (https://data.gats-inc.com/saber/custom/Temp_O3_H2O/v2.0/). The ERA5 reanalyses dataset available were retrieved from ECMWF's Meteorological Archival and Retrieval System (MARS), for

further details see (https://www.ecmwf.int/en/forecasts/datasets/browse-reanalysis-datasets), (Hersbach et al., 2020).



*Author contributions.* GB, DP, LP, HB, NB designed the methodology and performed the analysis. GB, DP, HB, LP, NB, JB, VA, LS contributed to the discussion of the results. GB, DP, LP and DB prepared the manuscript with contributions from all co-authors.

*Competing interests.* The authors declare that they have no conflict of interest.

*Acknowledgements.* This work is part of the MESO Project "Modelling and forecasting the secondary effects of the Antarctic ozone hole",
registered under no. 8887.130199/2017-00. The authors would like to thank the CAPES (Coordination of Improvement of Higher Education Personnel) and COFECUB (French Committee for the Evaluation of University Cooperation with Brazil) program responsible for promoting this research. The authors are also grateful for data provided by WOUDC, NASA, ECMWF and TIMED/SABER with daily mean the total column ozone with ground-based (Brewer Spectrophotometer) and satellite data (TOMS/OMI), meteorological daily mean data by ERA5 reanalysis, and vertical profile by SABER. To the National Institute for Space Research (CRS/INPE-MCTIC) for the support and availability
of surface data.



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



**Table 1.** Monthly climatological values, their standard deviations and $-1.5\sigma$ limit for August, September, October, and November for the Santa Maria station.

| Month | $O_3$ **Climatology in DU ($\mu$)** | **Standard Deviation in DU ($\sigma$)** | **Limit -1.5$\sigma$ in DU ($\mu - 1.5\sigma$)** |
|---|---|---|---|
| August | 290.4 | 12.1 | 272.2 |
| September | 296.8 | 9.4 | 282.7 |
| October | 289 | 11.8 | 271.3 |
| November | 285 | 8.9 | 271.6 |





**Table 2.** AOH influence events from 1979 to 2020, for SM. The table presents the day of the event in MMDDYYYY format, and ozone content reductions relative to climatology in %, and QBO phase.

| Event Day | $O_3$ Reduction % | QBO Phase | Event Day | $O_3$ Reduction % | QBO Phase | Event Day | $O_3$ Reduction % | QBO Phase |
|---|---|---|---|---|---|---|---|---|
| 09/27/1979 | 4,2 | Negative | 09/14/1997 | 14 | Positive | 08/08/2010 | 7,2 | Positive |
| 08/05/1982 | 11,4 | Positive | 10/16/1997 | 8,3 | Positive | 09/08/2010 | 5,4 | Positive |
| 09/09/1982 | 13,2 | Positive | 11/02/1997 | 6 | Positive | 13/13/2010 | 5,2 | Positive |
| 09/23/1982 | 6,2 | Positive | 11/19/1997 | 6,5 | Positive | 10/22/2010 | 10,1 | Positive |
| 10/08/1982 | 5 | Positive | 10/24/1998 | 10 | Negative | 09/05/2011 | 4,4 | Negative |
| 10/15/1982 | 7,3 | Positive | 08/21/1998 | 8,1 | Positive | 10/21/2011 | 4 | Negative |
| 10/21/1982 | 9,4 | Positive | 10/07/1999 | 4,7 | Positive | 09/14/2012 | 9,7 | Negative |
| 09/30/1983 | 7,8 | Negative | 11/03/1999 | 4 | Positive | 09/22/2012 | 6,5 | Negative |
| 10/14/1983 | 4 | Negative | 09/23/2000 | 8,9 | Negative | 10/14/2012 | 12,5 | Negative |
| 10/16/1984 | 9,2 | Negative | 10/12/2000 | 7,2 | Negative | 10/23/2013 | 13,6 | Positive |
| 11/05/2014 | 7,5 | Negative | 10/26/2000 | 6,7 | Negative | 08/10/2014 | 7,5 | Negative |
| 08/09/1985 | 7,3 | Positive | 08/15/2001 | 5,8 | Negative | 08/22/2014 | 12 | Negative |
| 08/23/1985 | 11 | Positive | 09/23/2001 | 7,7 | Negative | 10/13/2014 | 4,7 | Negative |
| 09/03/1985 | 16 | Positive | 08/18/2002 | 12,5 | Positive | 11/03/2014 | 4,3 | Negative |
| 10/12/1985 | 10,6 | Positive | 10/15/2003 | 10,6 | Negative | 09/22/2015 | 7,7 | Positive |
| 11/07/1985 | 7,6 | Positive | 08/22/2004 | 9,3 | Positive | 11/03/2015 | 8,3 | Positive |
| 11/05/1985 | 4 | Negative | 09/12/2004 | 6,8 | Positive | 08/25/2016 | 12,9 | Positive |
| 08/09/1987 | 7,6 | Positive | 10/03/2004 | 4,6 | Positive | 09/05/2016 | 10,5 | Positive |
| 10/16/1987 | 6,1 | Positive | 10/16/2004 | 12,3 | Positive | 09/12/2016 | 9,4 | Positive |
| 09/02/1988 | 5,7 | Negative | 09/29/2005 | 5,5 | Negative | 10/20/2016 | 22 | Positive |
| 08/24/1990 | 8,9 | Positive | 10/11/2005 | 5,2 | Negative | 08/26/2017 | 12,6 | Negative |
| 09/06/1990 | 16,8 | Positive | 11/16/2005 | 4,6 | Negative | 09/18/2017 | 8 | Negative |
| 09/16/1990 | 9,1 | Positive | 08/07/2006 | 10,9 | Positive | 11/11/2018 | 13,8 | Negative |
| 10/01/1990 | 4,2 | Positive | 08/23/2006 | 11,4 | Positive | 09/16/2019 | 14,8 | Positive |
| 10/09/1990 | 9,7 | Positive | 09/19/2006 | 6,6 | Positive | 08/28/2020 | 15,4 | Positive |
| 09/03/1992 | 8,5 | Positive | 10/07/2006 | 9,7 | Positive | 09/03/2020 | 17,7 | Positive |
| 09/21/1992 | 12,6 | Positive | 10/15/2006 | 9,1 | Positive | 10/21/2020 | 11 | Positive |
| 10/10/1992 | 7,5 | Positive | 11/17/2006 | 12,9 | Positive | 11/22/2020 | 10,7 | Positive |
| 08/15/1993 | 9 | Negative | 08/16/2007 | 6,1 | Negative | | | |
| 08/25/1993 | 8,5 | Negative | 09/13/2007 | 7,4 | Negative | | | |
| 10/19/1993 | 13,2 | Negative | 10/07/2007 | 10 | Negative | | | |
| 10/31/1993 | 16,8 | Negative | 09/28/2008 | 7,3 | Positive | | | |
| 08/16/1994 | 4,2 | Negative | 10/12/2008 | 8,3 | Positive | | | |
| 09/17/1994 | 6,1 | Negative | 10/26/2008 | 7,8 | Positive | | | |
| 10/01/1995 | 7,7 | Positive | 11/01/2008 | 10,5 | Positive | | | |
| 09/20/1996 | 17,4 | Negative | 09/02/2009 | 16,1 | Negative | | | |
| 08/18/1997 | 16,5 | Positive | 09/26/2009 | 8,9 | Negative | | | |



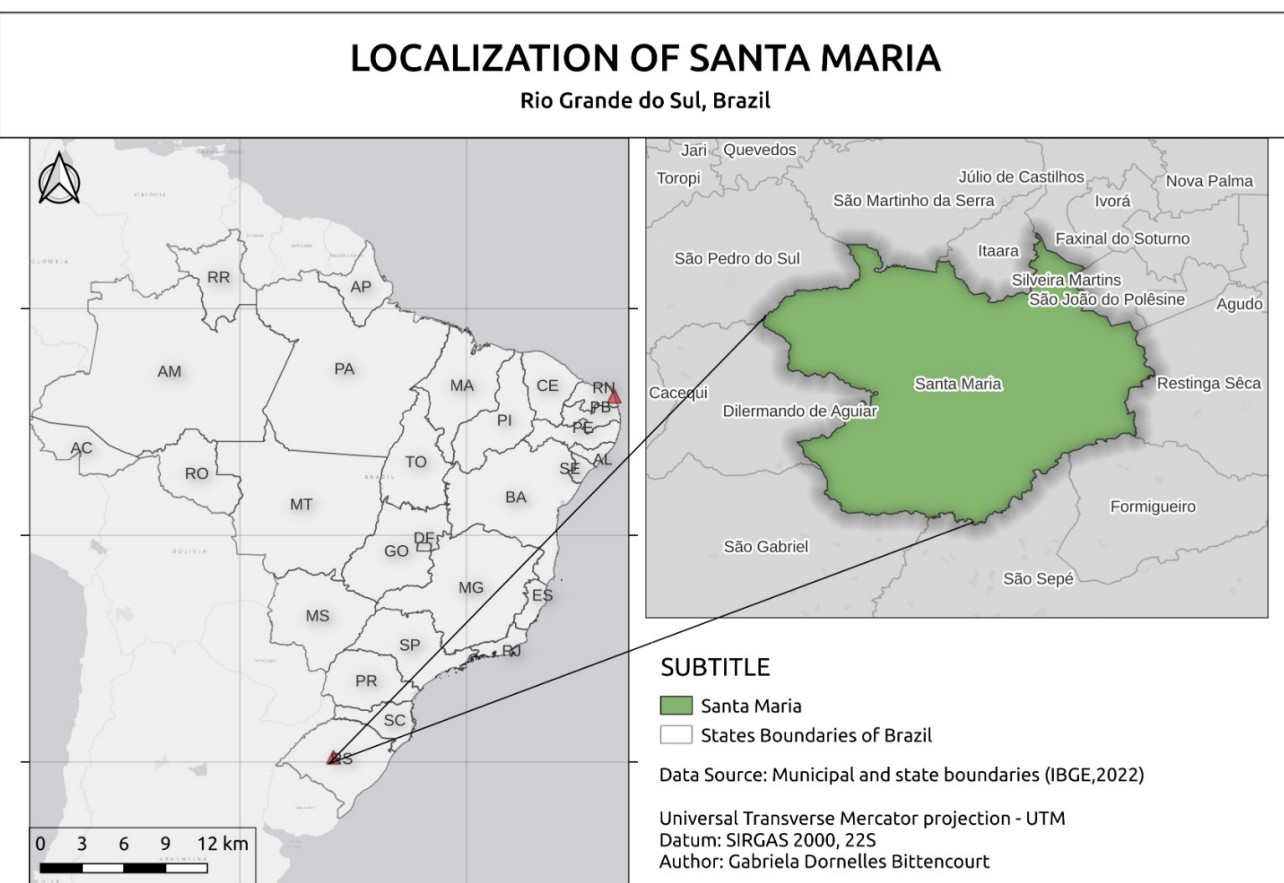

**Figure 1.** Study region used in this work. Santa Maria located in subtropical latitudes of Brazil (29.4ºS; 53.7ºW).



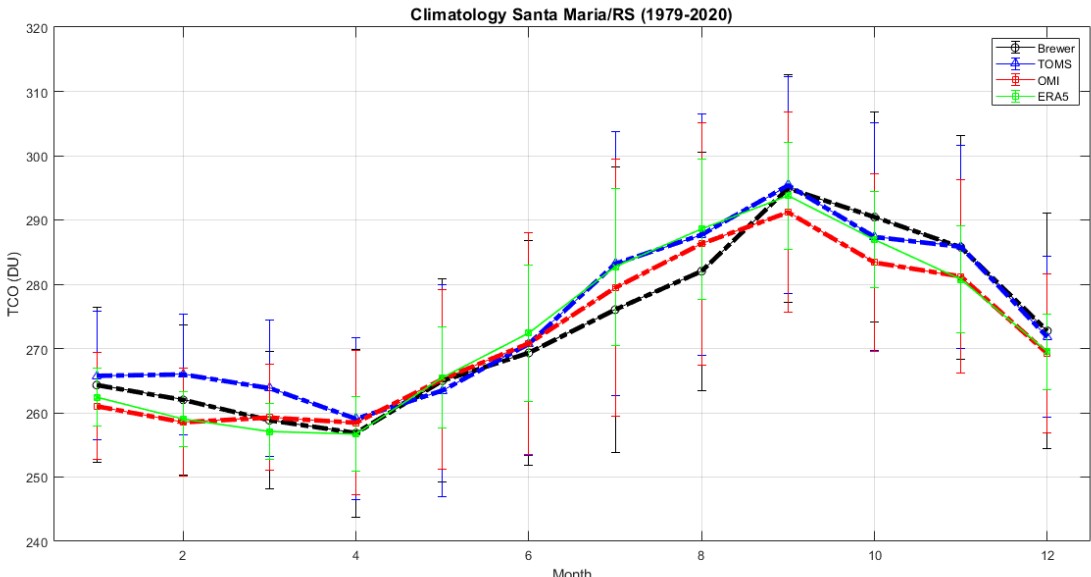

**Figure 2.** Monthly climatology for TCO in SM with satellite data TOMS (blue) and OMI (red) and ERA5 reanalysis (green) between 1979-2020 and, ground-based data with Brewer spectrophotometer (black) at intervals 1992-2017.





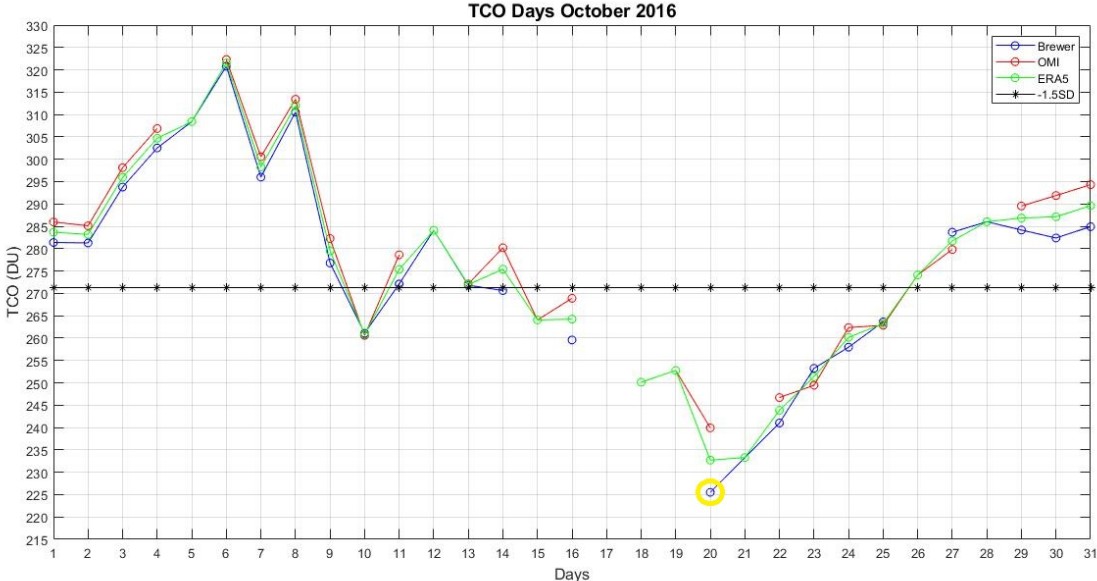

**Figure 3.** TCO values in October 2016 with data from Brewer (blue), OMI (red), and ERA5 (green). The limit of -1.5$\sigma$ in black line.





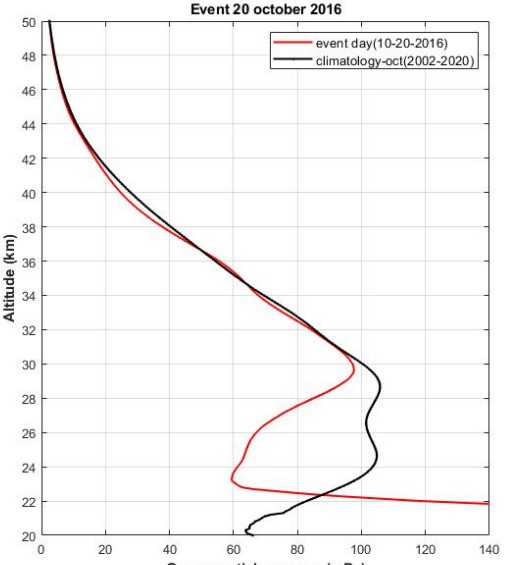
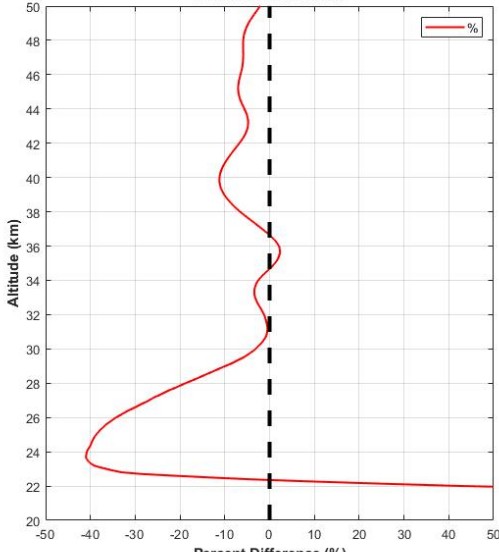

**Figure 4.** a) Vertical profile of $O_3$ from the SABER satellite for October 20, 2016 (in red) and climatology for the month of October (in black) in mPa. b) Percent difference in % (red line) on the day of the event, in relation to the weather for October.



**Figure 5.** Potential vorticity fields at 20 hPa of potential temperature for October 18-21, 2016.

(a) 18

(b) 19

(c) 20

(d) 21



**Figure 6.** Vertical section of the atmosphere between 1000 and 5 hPa for the days of the event in October 2016, with potential temperature (solid line) and jet stream (purple shading). Longitude set to 54ºW and Latitude SM represented in the purple triangle at 29°S.

(a) 18

(b) 19

(c) 20

(d) 21



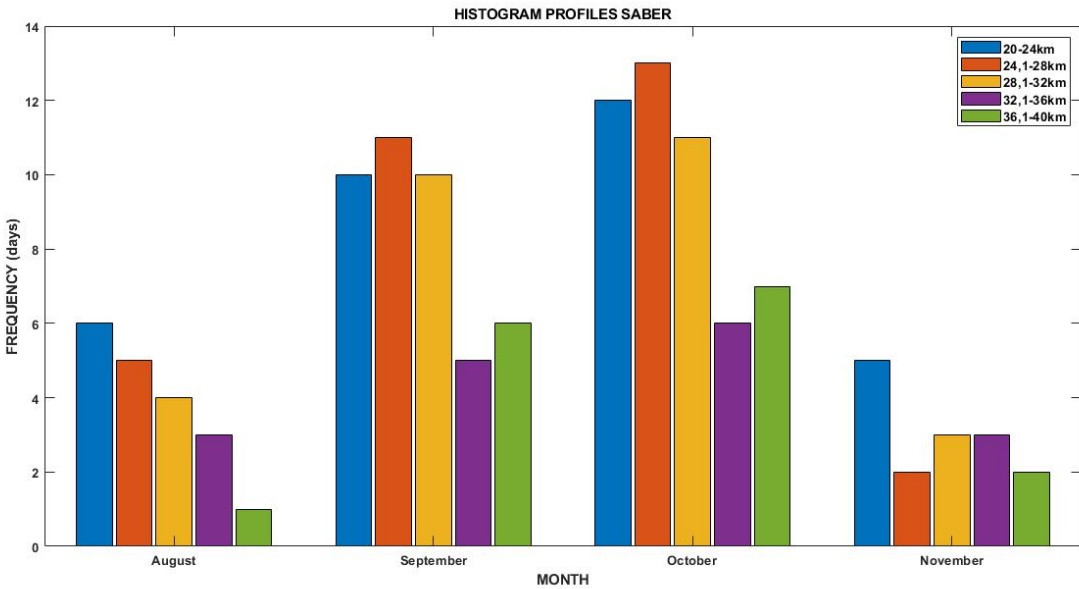

**Figure 7.** Histogram with the frequency of occurrence of events for each altitude group, using SABER satellite data from 2002-2020.



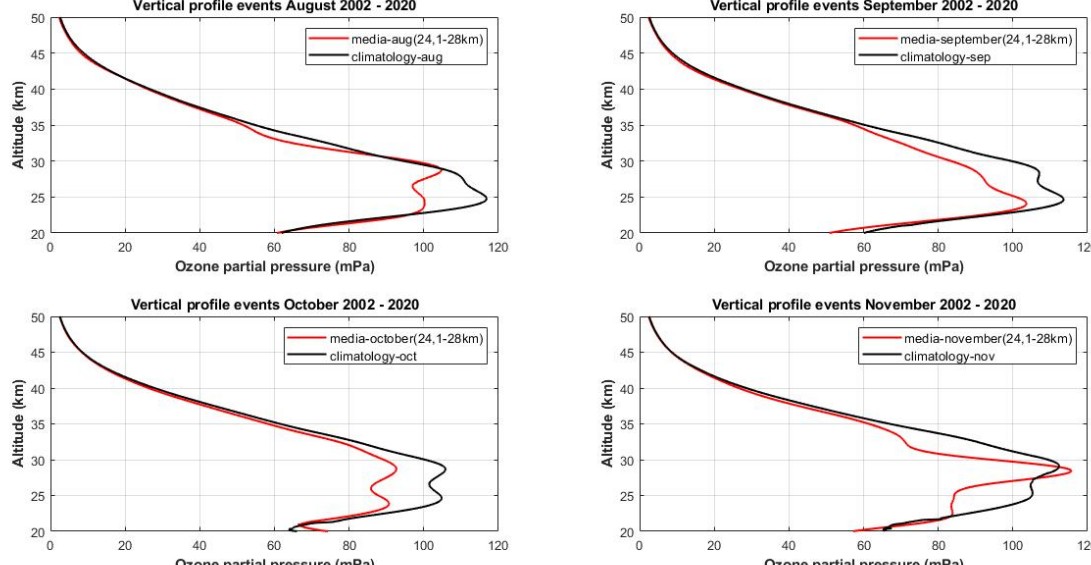

**Figure 8.** Monthly vertical profile between 2002-2020 in the layer 24.1-28 km altitude, with the average of events per month (a) August, b) September, c) October, d) November) in red and the monthly climatology in black, in partial pressure unit of $O_3$ (mPa).



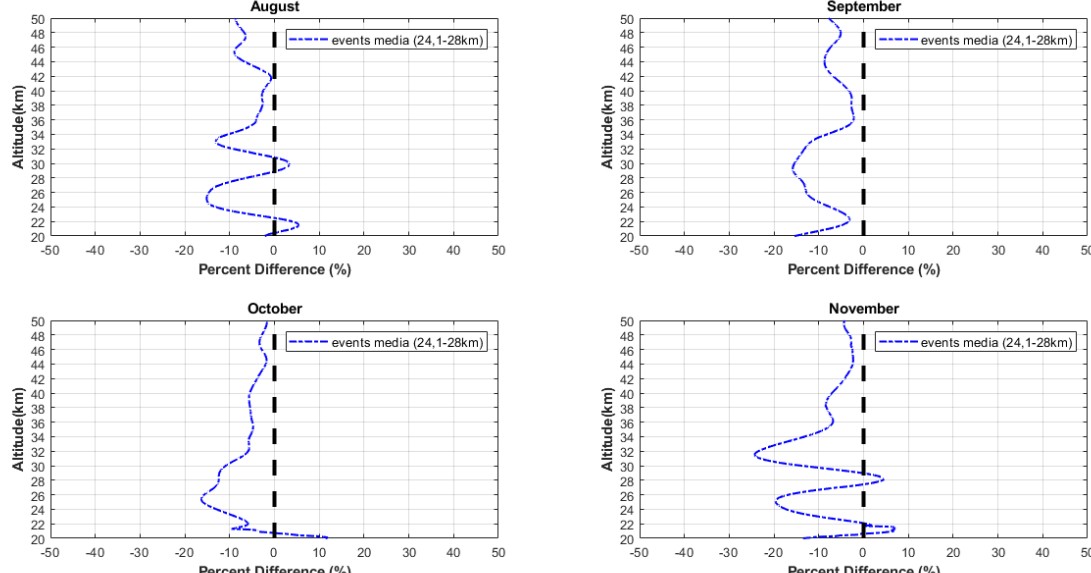

**Figure 9.** Monthly percentage difference in the altitude layer 24.1 – 28 km, in relation to the average of the AOH influence events per month, (a) August, b) September, c) October and d) November in the period from 2002 to 2020 with SABER data.



**Figure 10.** Monthly anomaly (a,b,c,d) of potential vorticity at 20 hPa of potential vorticity in the months of occurrence of the AOH influence events on SM from 1979 to 2020.

(a) August

(b) September

(c) October

(d) November





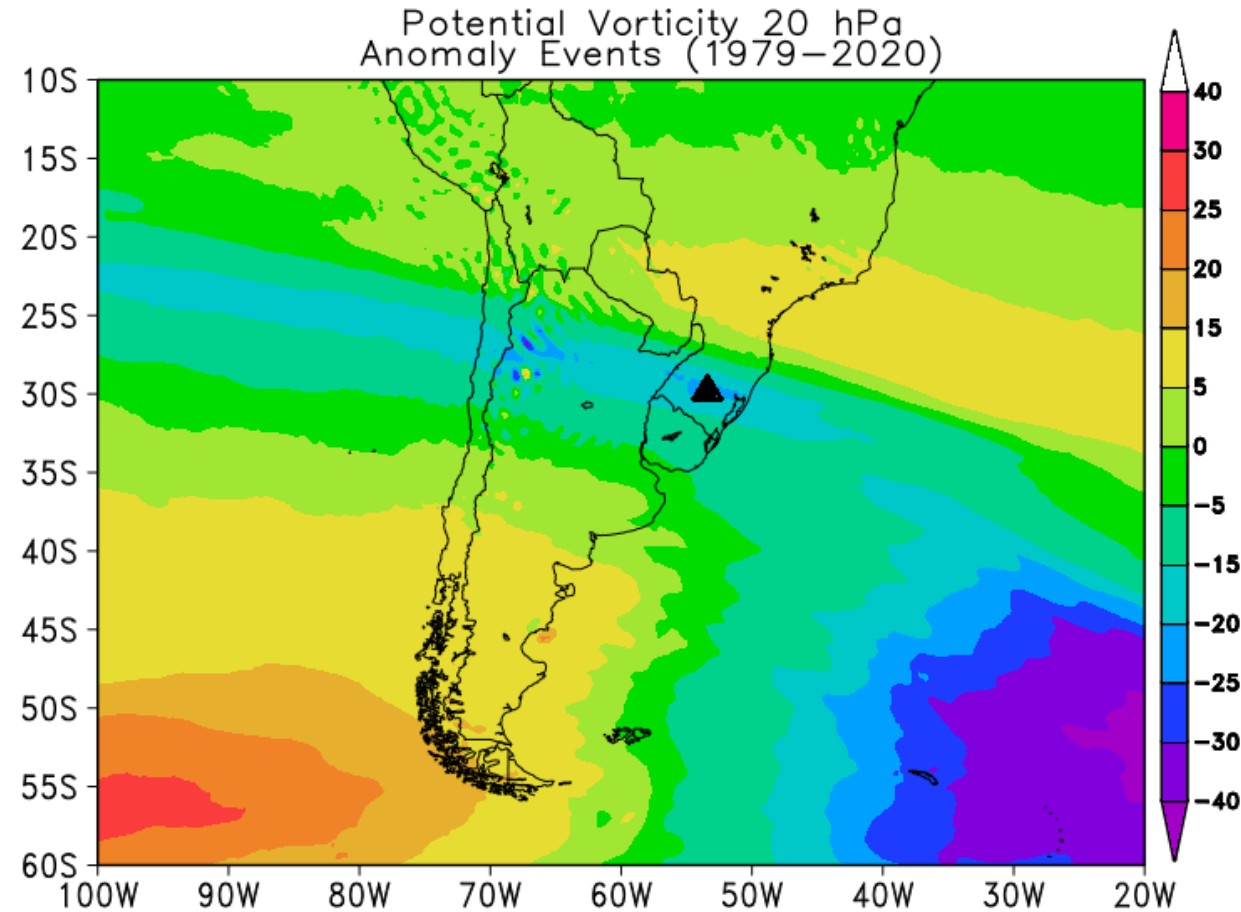

**Figure 11.** Potential vorticity field at 20 hPa showing the anomaly of the 102 events between 1979 to 2020.



**Figure 12.** Monthly climatology of the vertical section the atmosphere between 1979 to 2020 from August to November (a, b, c, and d). Layer of 1000 to 5 hPa potential temperature (solid line) and jet stream (purple shading). Longitude set to 54°W and latitude SM represented in the purple triangle at 29°S.

(a) August

(b) September

(c) October

(d) November





**Figure 13.** Average of the 102 AOH influence events identified during the period 1979 - 2020 of the vertical slice of the atmosphere between 1000 and 5 hPa, showing potential temperature (solid line) and jet stream (purple shading). Longitude SM set to 54°W and latitude represented in the purple triangle at 29°S.



**Figure 14.** Conceptual model showing the horizontal section (a) of the atmosphere with the position of the stratospheric and tropospheric jets at 20 hPa and 200 hPa, and the vertical section of the atmosphere (b) between 1000 and 5 hPa showing the coupling of the jets during the influence events of the AOH in mid-latitude regions.

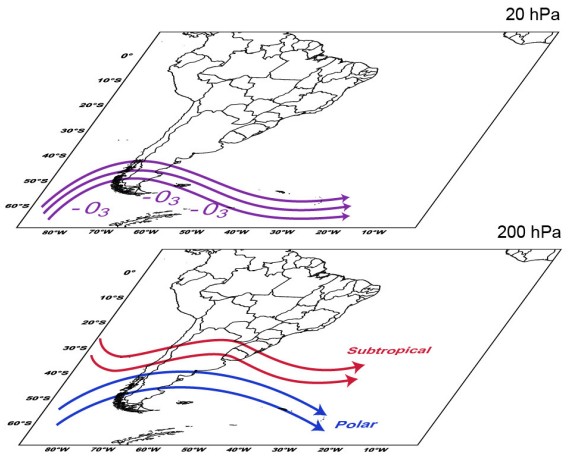

(a) Horizontal Section of Atmosphere

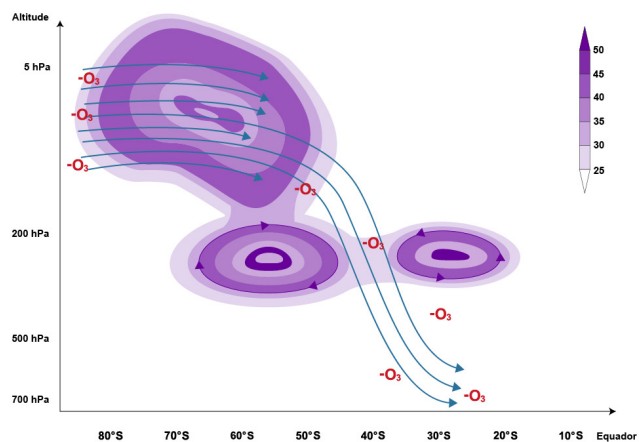

(b) Vertical Section of the Atmosphere