# Peer review of "Measurement report: Influence of the Antarctic Ozone Hole in Southern Brazil: Conceptual model for 42 years of analysis the atmospheric dynamics on ozone"

_EGUsphere, 2023_

## Author Comment (AC1)

**Response to Anonymous Referee #1 (RC1):**

We would like to thank the first referee for his time, valuable feedback and comments. Below are the original comments and the authors' response (in blue). The changes will be in the new version of the manuscript (in red).

##############################################################################

Review of Bittencourt et al., ***Measurement report: Influence of the Antarctic Ozone Hole in Southern Brazil: Conceptual model for 42 years of analysis the atmospheric dynamics on ozone***

**General Comments**

The subject of the submitted manuscript is one of great interest, episodes of low total ozone observed in the city of Santa Maria in Brazil located at 29 degrees latitude, caused by filaments of stratospheric air originating in the Antarctic ozone hole.

The authors identify low ozone events using ground-based and satellite-based measurements of total ozone that are 1.5 standard deviations below the climatology, and then class these as of polar influence if the event is associated with an increase in the magnitude of potential vorticity. A case study is then presented in some detail of a noteworthy event of 20 October 2016, which I note has already been the subject of previous publication by the authors. Vertical profiles of the low ozone episodes are studied using SABER data, and finally some analysis is presented of the broader dynamic situation, with a focus on the stratospheric and tropospheric jets.

Unfortunately, I believe the manuscript requires major revisions before it is suitable for publication.

It is not at all clear to me that there is anything new here compared to the authors' previous works on the same subject cited in the references (Bittencourt et. al 2018, Bittencourt et al. 2019, Bresciani et al. 2019, Peres et al. 2019). There is a large amount of overlap with these references.

Agreed. The references in question show results that have already been analyzed and developed in relation to analysis of the influence of the Antarctic Ozone Hole. The difference for this work is the database analyzed using more than 40 years of measurements available using different types of instruments, where it was possible to identify the occurrence of 102 events (as shown in table 2) influenced by AOH in the southern region of Brazil (Santa Maria). In this way, it was possible to create a conceptual model of the atmosphere during the occurrence of these events over the study region.

Therefore, firstly, the new findings of this study need to be made much more explicit.

Agreed. The authors agree that a rewriting of both the discussions and the conclusions identified in this work is necessary.

Secondly, the writing style in general needs to be made much easier for the reader to follow. Each section currently contains a large amount of background material and repetition before reaching the main point. I then often found it quite hard to find and understand the point being made, and what exactly was being said. I suggest shortening the background discussion and removing the repetition throughout the manuscript, and then using more text to explain your new findings more clearly.

Agreed. We will modify the writing of the text, making it more concise and fluid so that the reader has a good understanding.

**Specific comments**

Figure 10 is not described properly in the caption – from the text I think it shows the composite of 20 hPa PV for all low ozone events with low PV for each month? I think it would also be interesting to show the composite for low ozone events with high PV for comparison.

As the focus of the analyzes is during the AOH activity period (August to November), the analyzes presented here are in relation to these four specific months. Figure 10 presents the monthly anomaly in relation to the climatology of each month, for the 42 years of analysis (1979 to 2020). An organization in writing the new version of the manuscript will be made.

Figure 12 The figure caption says it is the 'monthly climatology', not just for low ozone events – is that correct? Figure 13 is very similar to figure 12 so it's hard to see that there is any difference in the position of the jets when there is low ozone event compared to the average situation.

Agreed. Figure 12 presents the monthly climatology of the vertical cut of the atmosphere, between 1000 and 5 hPa only for the months of AOH activity, that is, from August to November, for the period from 1979 to 2020. The authors agree that perhaps a better description of these analyzes is necessary to make it clear that the study is around four months of AOH activity over the southern hemisphere. In figure 13, the analysis is only for the 102 events of temporary decrease in ozone content identified over Santa Maria.

Figure 13 doesn't look very similar to Figure 6 though, so does this mean the event in figure 6 doesn't show the usual pattern for low ozone events?

As mentioned in the comment above, figure 13 represents only the average of all identified events (table 2) during 1979 - 2020. Figure 6 only shows the behavior of the stratospheric and tropospheric jets during the days of the selected event, in this case October 20, 2016. Below, another example of an event identified in the region.

I am very confused about your "conceptual model", and I think it needs to be explained much more clearly. Figure 14 doesn't look at all like figure 13.

The authors agree that the discussions need to be clearer.

Figure 14 appears to show ozone-poor air from inside the polar vortex moving across the width of the stratospheric polar jet (why would it do that?) then moving downwards between the two tropospheric jets to finish close to the surface at 700 hPa. This does not seem to have relevance to the rest of the study, because in the text, all the discussion has been about ozone at altitudes above 20 km.

Figure 14 has been reformulated for better understanding by the reader. The objective of this conceptual model is to show how this transport of ozone-poor air masses reaches mid-latitude regions during the active period of the AOH. Initially, the destabilization of the vortex releases masses of air from within, where the lowest concentrations of O3 are found. With the help of tropospheric jets (subtropical and polar jets), this transport occurs from high-altitude regions to mid-latitude regions. This idealized movement considers the entire period of TCO data analyzed over the Santa Maria region, after identifying these

AOH-influenced events, with more than 40 years of analysis. The blue lines indicate this air mass movement.

[Figure]

Regarding the discussion of the QBO (lines 320-326), you need to show by a simple test that the difference between the number of events in the different phases is statistically significant.

Agreed. The analyzes presented in this paper regarding QBO were very preliminary results, only relating the QBO phase to the AOH period and the identified events. However, this is not one of the focuses of the paper, perhaps carrying out a more in-depth analysis in another work would be more interesting.

Please follow the style guide (https://www.atmospheric-chemistry-and-physics.net/submission.html#english) for the format of dates.

Thank you very much for the suggestions, date corrections will be made.

###

---

## Author Comment (AC2)

Preprint egusphere-2023-1471, RC2

**Response to Anonymous Referee #2 (RC2):**

We would like to thank the first referee for his time, valuable feedback and comments. Below are the original comments and the authors' response (in blue). The changes will be in the new version of the manuscript (in red).

###############################################################################

This paper examines events where low ozone is observed over Southern Brazil over the last ~42 years and identifies cases where such events can be attributed to transport of ozone-poor air from the Antarctic polar vortex during ozone hole season. The analysis centers on observations of total column ozone from a combination of ground-based and spaceborne sensors, supplemented by vertically resolved observations from TIMED SABER during the most recent two decades. ERA-5 reanalysis fields are used to diagnose the transport associated with these events and statistics are compiled.

Unfortunately, this paper needs a lot of work before it can be ready for publication. The discussion is hard to follow in many places. In particular, the manuscript spends a lot of time talking about findings of prior studies by the authors and others, and it is very challenging for a reader to discern what about this study is "new". Is it the addition of SABER data? The conceptual model? The inclusion of more years? I don't think it can be the application of the statistical analysis as that was covered in earlier papers. It would be best to have a clear discussion in the introduction of what was done/found previously and then briefly preview the new findings (or at least the new avenues of inquiry).

Agreed. The authors agree that there is a great extension in the writing regarding previous findings. These discussions will be revised and better written, summarizing only the most important information from the document. The focus of this document is on fact the idealization of a conceptual model of the atmosphere in relation to the occurrence of events influenced by the Antarctic Ozone Hole (AOH) over southern regions of South America. Furthermore, the document provides an analysis of more than 42 years of total ozone column over these regions that suffer indirect influences from AOH during its active period. With all this, the manuscript will be redesigned by the authors for a better understanding by the reader of the topic covered and its main points.

I have a list of specific concerns detailed below. Foremost among them is my serious questions about the author's "conceptual model", at least as described in Figure 14. Also, I am not convinced by authors argument that the "connection" of the various wind jets is key to the transport processes they describe. Such "connections" may indeed be important, but I do not believe the authors have provided sufficient evidence to demonstrate that. At best they have shown a correlation (though not quantified its statistical significance), but are there other such connection events that are not associated with changes in ozone? Similarly, I am totally unclear what is meant by "funneling" in this context.

The authors thank you for your comments. The main idea of the manuscript, as described above, was to show the idealization of a conceptual model during AOH influence events over mid-latitude regions, based on a database of more than 42 years. Describing the dynamic behavior of the atmosphere during these thinning events through the analysis of the jets showed great evidence that this low ozone content reaches mid-latitudes through a "coupling" of the two systems (stratospheric jet and tropospheric jets). The events of decrease in ozone content, in this case, were analyzed only during the active period of the AOH, between August and November between 1979 and 2020.

Finally, I'm afraid the standard of writing in this paper is rather poor, with inappropriate word choices and incorrect grammar throughout. I have highlighted some places where corrections are needed. However, this is far from a comprehensive list. Until this paper has received significant attention from a native English speaker and/or a copy editor, I'm afraid it will be unsuitable for publication on those grounds alone. That said, that alone is insufficient. The authors need to work hard to structure this paper more clearly, with the old and new material delineated, and the "case study" separated from the statistical work.

Agreed. The document will undergo a more detailed technical analysis to allow for more fluid reading and easier understanding of the objectives, results and conclusions identified.

**Specific comments:**

Line 13: The sentence "In the dynamic analysis…" is very unclear. Do you mean something like "An analysis of the dynamics associated with these events found an increase in absolute potential vorticity…" (do we really need to call out the vertical and horizontal cross sections?).

Yes. In this case, the idea is to say that the stratospheric dynamics, based on the potential vorticity fields, showed an increase in their values (APV) during the identified events that occurred at their largest, as expected, in September and October. The sentence will be better written for better understanding.

Line 14: "The conceptual model" - this is the first time you're introducing it, so you should say "Our conceptual model". Also, again suggesting that this is a purely horizontal and vertical conceptual model feels odd here. Your conceptual model is for the whole atmosphere, you're just choosing to describe it with reference to those cross sections, but that's a detail in my view.

Agreed. Thanks for the suggestion.

Line 15: "jet" -> "jets" I believe.

Agreed.

Line 16: "O3 content in Santa Maria" -> "Total ozone column above Santa Maria" surely, we're not talking about the air in the city at ground level.

Agreed.

Line 17/18: "medium and high levels in the atmosphere" is unacceptably vague. To some (e.g., space physicists) the "upper troposphere" would be considered "low", to others (e.g., boundary layer meteorologists) it would be considered "high". Be specific (e.g., "mid-stratosphere") or better still give approximate altitude ranges.

Agreed. Thanks for the suggestion. The text will be reformulated to better identify the time at which it is being analyzed.

Line 20: I really don't know what is meant by "through the energy balance of the planet" in this context. It feels like it's stuck in the middle of a bunch of points about life on

Earth. I suggest you delete it or reword the whole sentence for better organization and clarity.

Thanks for the suggestion.

Line 23: "most important" really!!!? Surely many would put oxygen higher on the list (or CO2 for that matter). Important yes, but you really can't say "most important".

Agreed. Thanks for your consideration, we rewrite the sentence… "In 1840, when scientists discovered the gas ozone ($O_3$), studies show that **is an important** trace gas for sustaining life on Earth, due to its ability to absorb ultraviolet (UV) radiation incident on the atmosphere…".

Line 24: "About 90%"? I presume by this you mean that 90% of the atmospheric ozone is found in the stratosphere, but the way you've written it, it sounds like you're claiming a 90% mixing ratio in the stratosphere. Please reword.

Agreed. Thanks for the suggestion.

Line 26: This is completely wrong. The BDC is relevant for the ozone hole yes, but it is highly misleading to cite dynamics as the fundamental cause of the ozone hole. Please instead talk about the role of chlorine and polar stratospheric clouds and cite the paper by Solomon (and/or others) as you do later in the text.

Thanks for the suggestion.

Lines 30/31: Please clarify the vertical region over which this barrier applies (e.g., "from the upper troposphere and into the mesosphere" or wherever it is the case).

This barrier is nothing more than the polar vortex in the stratospheric layer. This isolation occurs due to the strong winds that surround the polar regions (both in the Arctic and Antarctica), which intensifies during the winter due to the low temperatures in the stratosphere. This barrier formed by the vortex prevents the mixing of air from outside to inside and vice versa. Vertically this barrier extends from the tropopause region to the mid-stratosphere.

Line 33: "poor ozone" -> "ozone-poor"

Agreed.

Line 42: "according to" -> "as shown in"

Agreed.

Line 43: "figure 1" -> "Figure 1" for consistency with other parts of the paper (and with ACP standards I expect).

Agreed.

Line 41/42: A badly worded sentence, please rewrite.

Agreed.

Line 45: Delete comma after "radiation"

Agreed.

Line 50: Reword this sentence. As it is you are stating that the sun is made of spectrophotometers.

Agreed. Thanks for the suggestion.

Line 53: Delete comma after "called"

Agreed.

Line 54: Move "to infer TCO" after "with a resolution of 0.5 nm"

Agreed.

Line 55: "This is made to prevent most of…" -> "This step is taken in order to reduce the majority of cloud interference…"

Agreed.

Line 67: "TOM'S" -> "TOMS"

Agreed.

Line 76: OMI is on the Aura satellite, not ERS-2!! (Also, not it's Aura, not AURA)

Agreed.

Line 82: See 76 (plus this is partly redundant with that earlier sentence)

Agreed. The repeated phrase will be removed.

Line 85: By "bands" do you mean spectral bands (in which case "channels") might be a better word, or view directions, in which case something like "pixels" might be better.

Agreed.

Line 103: Please explain whether (1) this interpolation is done by the SABER team or you and (2) why on Earth you felt the need to interpolate to such a fine resolution? This is way finer than any true information obtained by the SABER instrument. What difference did you find that made you pick this rather than sticking with the native SABER resolution or just using 1km?

The choice of resolution was simply to visualize in more detail all the information in the vertical analysis of ozone content, mainly during events influenced by AOH, where the objective is to identify the preferred heights at which this decrease occurs.

Section 2.1.3. You should note that ERA-5 assimilates ozone column (from Aura OMI I believe) and ozone profile from Aura MLS - this will affect the ERA-5 ozone product, giving more realistic results following the 2004 Aura launch.

Preprint egusphere-2023-1471, RC2

Agreed.

Line 129-133: A particularly clumsily worded pair of sentences.

Agreed. The sentence will be reformulated.

Section 2.2: This section heading needs major rewording. Firstly, it should be "Criterion" (or Criteria if there is more than one of them). I suggest "Criteria for defining Antarctic ozone hole influence on events".

Agreed. In this case, more than one criterion is analyzed to identify possible events influencing the AOH.. "Criteria for defining Antarctic Ozone Hole events"

Line 135/136: This sentence is garbled. Please clarify.

Also, you need to make it clear whether the standard deviations are computed separately for each month (e.g., separate numbers for September 2020, September 2021, etc.), or for all the months together (i.e., computing the standard deviation for all the September days in the 42-year window). Not until you show us Figure 2 does it become somewhat clear that it is the latter, but I'm still not sure even now.

Agreed, the sentence is meaningless the way it was written. Regarding the criteria for identifying possible days of AOH influence events, days are observed in which there is a temporary decrease in $O_3$ content in relation to the monthly climatology for the 42 years. In this case, the criterion uses the identification methodology presented by Peres (2016), where days are sought in which the daily average value of the total ozone column is lower than the climatological average minus 1.5 of the standard deviation of the same month. Both the calculation of the climatological mean and the standard deviation are made in relation to the entire period of analyzed data, that is, 42 years of TCO climatology for Santa Maria, they are used here in this first filter to identify possible days in which the content of $O_3$ temporarily decreased in the region.

The sentence between lines 135/136 was confusing, but below the explanation of the analysis criteria becomes clearer. However, a better description of this paragraph will be made.

Line 159: I think you mean the annual "cycle" not "variability", right? Even then I would tone it down and change "stands out" to "is clear."

Agreed. Yes, the "annual cycle is clear..." is better in the sentence. Thanks for the suggestion.

Line 161: You already defined the BDC (but then again, you'll probably be dropping that discussion earlier, so perhaps keep it here).

Agreed.

Line 171/172: This sentence is a pure assertion with no evidence given to back it up. Please work on a justification. In any case, change "Despite not being such a low value" to "Despite not being a particularly weak correlation."

Agreed. The entire paragraph has been reformulated for better clarity of information. As the figures relating to these analyzes were not added to the manuscript, only different

examples of these analyzes are referenced. If reviewers deem it necessary, figures can be added as extra material.

Line 175: "unlike" -> "in contrast with".

Agreed. Thanks for the suggestion.

Line 175: "at high levels of the atmosphere" is way too vague (as before). Be specific about the region (e.g., upper stratosphere?) and/or quote altitudes.

Agreed. Thanks for the suggestion and corrections will be made throughout the manuscript.

Line 177: Not sure about "as a function of" do you mean more like "driven by"?

Yes, "driven by" is better in the sentence.

Line 201 (and elsewhere): Why do you use the term "secondary"? Is it to distinguish it from the ozone hole itself. If so, I find it confusing, as I keep thinking you're referring to a double dip in the ozone over your site. I suggest dropping this terminology.

Yes, the term "secondary" was first used by Kirchoff et al. (1996) when they identified events of decreasing $O_3$ content that influenced mid-latitude regions. As the "primary" effect, is the action of the ozone hole in the Antarctic region, the term "secondary" refers to these "ejections of ozone-poor air mass from the Antarctic region to other latitudes of the globe, such as latitudes averages, being the case of the study region of this manuscript.

In this manuscript we will use the term **influence** to characterize secondary effect events of AOH on the region and SM.

Line 202: I suggest breaking the sentence after the Bresciani reference. Have a period, then drop the "after".

Agreed.

Line 205: "AURA" -> "Aura". But Aura is not an instrument it is a mission. Please name the Aura instrument (MLS? HIRDLS? OMI?) to be parallel with your reference to SABER rather than TIMED.

Agreed. Thanks for the correction.

Line 207: "BREWER" -> "Brewer"

Agreed.

Line 221: What do you mean by flaws in the data? If it's just the gaps then say gaps (or don't bother with this thought, you only mentioned it a few sentences ago). If you mean something else, then be more specific.

Agreed. Thanks for the suggestion. Mainly in relation to surface instruments, some periods are without data, due to technical problems with the equipment. Gap is better described in this case.

Preprint egusphere-2023-1471, RC2

Line 227 (and associated figures): I think you'd be far better off showing volume mixing ratio rather than partial pressure as it is conserved following air motions, which is the story you're attempting to tell. That is the unit that atmospheric chemists prefer.

Agreed. Thanks for the suggestion.

Line 234: "high values of reduction" is confusing. "Large reductions" would be clearer.

Agreed. Thanks for the suggestion.

Section 3.3: The start of this section reads like you're still in the case study, in which case it should be 3.2.2 (indeed, you should not have a 3.2.1 if you don't have a 3.2.2). But then you change to talk about the statistics. I suggest you rearrange/split this section to separate the remnants of the case study from the statistics discussion.

Agreed. Section 3.3 is still about the case study, so it is better to indicate it as 3.2.2, as you suggest. All statistical analyzes are described in section 3.4, which will now be 3.3.

Line 277: "Figure ??" - insert correct cross reference.

Agreed. The Figure in question is Figure 6.

Line 280: "These large-scale systems help to understand…" poor wording, these systems cannot help anyone understand anything (it's like saying "Thursday can help to understand"), what is it about these systems that you are using to illustrate our understanding.

Agreed. The sentence is lost in the discussion. Regarding this analysis, the objective is to show that these large-scale systems, such as stratospheric and tropospheric jets, "can" explain how atmospheric dynamics act during these events of decreasing O3 content.

Line 320 and associated discussion: Please verify whether these differences are statistically significant.

Agreed. In this manuscript, these statistics end up being information with few details, more studies need to be carried out to quantify and better understand this relationship (or anti-relationship) with the Quasi-Biennial Oscillation (QBO). As it deviates from the entire objective of the article, this discussion will be removed and will be worked on in another manuscript in the future.

Line 335: I would not say "lowest occurrence" - it's only lowest among the months you've chosen. Presumably April is lower still, right?

Agreed. For the analysis period (August to November), September and October are the months that present the most indications of event occurrence, while August and November, in that order, present fewer events during the period.

Line 338: "most intermediate", again, be specific.

Agreed. Corrections will be made in the new version of the manuscript.

Line 344: "it was established to analyze" - poor wording.

Preprint egusphere-2023-1471, RC2

Agreed. Corrections will be made in the new version of the manuscript.

Lines 347/348: The sentence "Regarding the climatology…" is confusingly worded. It sounds like you're saying that the climatology stands out against the climatology.

Agreed. Corrections will be made in the new version of the manuscript.

Lines 364/365: This description is very vague. How precisely is this calculation performed. Is it for all the days outside your 1.5-sigma threshold, or some fixed-length window around a central day. Is it a straight composite thereafter? More detail is needed.

Lines 366-368: Badly worded sentence. You refer to Figures 10b and 10c as if they tell us that September and October stand out with the most events, but the figures do not show that. Also, you should put the pointer to the figures in parentheses or have a comma after 10c in this case.

Agreed. Corrections will be made in the new version of the manuscript. In this case, figures 10b and 10c are referenced because an increase in the values of negative potential vorticity anomalies is observed, during the analysis period 1979 to 2020, stand out in the months of September and October. Firstly, analyzing table 2, these two months are the ones with the highest number of AOH influence events over mid-latitude regions.

Line 383: I have no idea what "isentropic ones" means in this case. Nor what you mean by "funneling". You need to point to specific feature in the plot. Are the contours getting closer? (Not really to my eye). This is your key point (or at least I think you want it to be) and yet I'm struggling to see what you see. Help the reader out!

Analyzes on isentropic surfaces allow identifying and monitoring the most relevant atmospheric characteristics in the analysis of atmospheric dynamics, as is the case in this work. Isentropic surfaces undergo vertical deflections of several kilometers and anomalous concentrations of stratospheric species can be observed in the lower troposphere, as is the case with ozone content. All analyzes referring to isentropic surfaces will be rewritten for better clarity for the reader.

Lines 383 and 395. The Santos and Rodriguez citations should not be in parentheses (use \citet rather than \citep in LaTeX).

Agreed. Corrections will be made in the new version of the manuscript.

Line 398: Negative temperature????!!! (do you mean negative anomaly?)

In this sentence the objective was to say that during the southern winter season, the polar vortex region presents negative temperatures, which ends up forming the "barrier", mentioned in previous comments. However, this is more of the same, so the authors suggest removing this phrase from the new version of the manuscript.

Line 402: Medium atmospheric levels again - be specific.

Agreed. Corrections will be made in the new version of the manuscript.

Line 404: Again, please tell me where I am supposed to see "isentropic coupling" - what is coupled to what in what figure?

Preprint egusphere-2023-1471, RC2

The sentence was rewritten in the new version of the manuscript.

Agreed. Corrections will be made in the new version of the manuscript.

Line 406 to 407: I am far from convinced by this argument. You need to compute trajectories or similar. Also, as discussed, you have not shown (unless I missed it) that there is no other such "coupling" events that do not result in low ozone. Until you show that, what you have shown could just be coincidence. In any case, correlation does not imply causation (you need to demonstrate that this is not the "post hoc ergo propter hoc" fallacy).

Agreed. Corrections will be made in the new version of the manuscript. The objective is to show how atmospheric dynamics, through jets, behave during AOH-influenced events that reach mid-latitude regions such as Santa Maria. The extensive database shows a consistency of these events in the region, and even so, each of the 102 presents a different behavior from the other, more details about the events can be analyzed in my doctoral thesis (Bittencourt, 2022).

Line 412: Again, I don't know what you mean by funneling. In any case "funneling of isentropic <what>". Do you mean funneling of the isentropes? I still don't really get what that means.

Corrections were made in the new version of the manuscript.

Line 413: "As shown"? Not really, at best it is "as illustrated" or (as you draw it). Also, see my notes on Figure 14.

Agreed.

Conclusions - I held short of commenting on these, pending all the work described above that is needed to verify that the conclusions you are drawing can be drawn.

Agreed. After all the corrections were made, the conclusion was rewritten, highlighting the important points of the work.

Figure 2: I'd suggest you artificially stagger the different datasets by a few days so the error bars can be more readily distinguished (make a note in the caption that you've done so).

Figure 4: As discussed, I urge you to plot in this in vmr, perhaps as a function of pressure (or better yet, potential temperature) rather than altitude. You do not comment on the huge anomaly below 22km. It's possible this is dynamical and will go away when plotted in vmr (another reason to do so). Otherwise, please explain this, as it's far more remarkable than the feature you're focusing on.

According to the suggestions, all figures that analyze vertical profile will be in ozone mixture ratio (ppmv). Below is figure 4 with the corrections made. These differences below 22 km can in fact be explained by the dynamic influences of the region, and also by the instabilities that the satellite presents at the beginning of the measurements.

[Figure]

[Figure]

Figure 5: My understanding is that it is far better to plot PV on a potential temperature surface. It changes rapidly in the vertical and the features you show may reflect that rather than true dynamics.

Agreed. All analysis involving the potential vorticity were corrected for potential temperature levels and will be added to the new version of the manuscript.

Figure 6: By "jet stream" in the caption do you mean winds? If so, say so? Is it just zonal wind? If so, say so. What are the thin lines with arrows on? Are they some measure of stream function? What does their density signify? Why are they not densest where the jets are strongest? Last sentence of the caption is completely unintelligible to me. Is it supposed to talking about where your station is?

The thin lines indicate wind flows, they will be removed so that there is no confusion regarding what we want to show with the jet streams. Regarding the last sentence of the caption, I wanted to make the measuring station very clear, in this case Santa Maria. As the figure is a vertical slice, this slice is made at the station's longitude (~54⁰ west), and the purple triangle on the x-axis of the figure represents the station's latitude (~29⁰ south). The caption will be rewritten.

Figure 8: Again, plot in vmr rather than partial pressure.

All figures, regarding the vertical profile, were redone in ppmv.

[Figure]

Figure 10: Again, you should plot this on a potential temperature surface. Also give more detail in the caption and/or the text on how this anomaly composite (if that's what it is) is computed.

Agreed. The anomaly calculation is made using the average of all events of each month, in this case the 102 events identified during the period from 1979 to 2020, and then this value is subtracted from the climatological average of the month, here the months of analysis are August to November.

Figure 12: See comments on Figure 6

Corrections will be made in the new version of the manuscript.

Figure 13: As figure 12.

Agreed. Corrections will be made in the new version of the manuscript.

Figure 14: What do the lines with arrows mean in both plots? For 14b, are they supposed to denote some kind of motion (so the x-axis is time as much as it is latitude?). If so, this is completely unbelievable. I do not for a moment think that ozone can be transported from ~10hPa down to ~500hPa, as you imply, over this latitude range or any kind of vortex transport timescale. Your work refers only to stratospheric phenomena, this diagram describes a completely different (and in my view unrealistic) process.

In relation to the lines in figure 14a, at 20 hPa the position of the stratospheric jet (polar vortex) is shown, indicating little ozone content inside the vortex, for the active period of the AOH (August to November). At 200 hPa the position of the tropospheric jets is presented, subtropical in red lines, and the polar jet in blue lines. As these figures are only representative of the position of these jets, the authors suggest removing this figure, as it only shows more than what is already available in the literature. The focus of figure 14 would then be the second part (14b). The blue lines indicate the movement of the air mass, leaving the stratospheric region at high latitudes, where there is a decrease in O3 during AOH activity, heading towards the mid-latitude regions, through the joint movement of the two tropospheric jets. Figure 14b has been reorganized, where a vertical cut is made in the atmosphere highlighting the movement of this ozone-poor air mass. The x-axis only indicates the latitudes, it has nothing to do with the time in which

this air mass arrives in these regions from the moment it moves from the Antarctic region, where the AOH is.